# Who You Are Matters: Bridging Topics and Social Roles via LLM-Enhanced Logical Recommendation

Qing Yu[1,*] Xiaobei Wang[2], Shuchang Liu[2], Yandong Bai[2], Xiaoyu Yang[2], Xueliang Wang[2], Chang Meng[2], Shanshan Wu[2], Hailan Yang[2], Bin Wen[2], Huihui Xiao[2] , Xiang Li[2] , Fan Yang[2], Xiaoqiang Feng[2], Lantao Hu[2], Han Li[2], Kun Gai[2], Lixin Zou[1,†]

[1] Wuhan University [2] Kuaishou Technology

{yu_qing, zoulixin}@whu.edu.cn,
{wangxiaobei03,liushuchang,chengfeng05,yangxiaoyu,wangxueliang03,mengchang,
wushanshan03,yanghailan,wenbin,xiaohuihui, lixiang44,yangfan,fengxiaoqiang,
hulantao,lihan08}@kuaishou.com, gai.kun@qq.com

## Abstract

Recommender systems filter contents/items valuable to users by inferring preferences from user features and historical behaviors. Mainstream approaches follow the learning-to-rank paradigm, which focuses on discovering and modeling item topics (e.g., categories) and capturing user preferences for these topics based on historical interactions. However, this paradigm often neglects the modeling of user characteristics and their social roles, which are logical confounders influencing the correlated interests and user preference transition. To bridge this gap, we introduce the *user role identification task* and the *behavioral logic modeling task* that aim to explicitly model user roles and learn the logical relations between item topics and user social roles. We show that it is possible to explicitly solve these tasks through an efficient integration framework of Large Language Model (LLM) and recommendation systems, for which we propose *TagCF*. On the one hand, *TagCF* exploits the (Multi-modal) LLM's world knowledge and logic inference ability to extract realistic tag-based virtual logic graphs that reveal dynamic and expressive knowledge of users, refining our understanding of user behaviors. On the other hand, *TagCF* presents empirically effective integration modules that take advantage of the extracted tag-logic information, augmenting the recommendation performance. We conduct both online experiments with an industrial environment and offline experiments on public datasets to verify *TagCF*'s effectiveness, and we empirically show that the user role modeling strategy is potentially a better choice than the modeling of item topics. Additionally, we provide evidence that the extracted logic graphs are empirically a general and transferable knowledge that can benefit a wide range of recommendation tasks. Our code is available in https://github.com/Code2Q/TagCF.

## 1 Introduction

Recommender systems have become an indispensable tool to mitigate information overload and are commonly employed on various online platforms, from e-commerce to video streaming, assisting users in finding personalized content. Traditional recommendation systems [43, 55, 24] typically adhere to the learning-to-rank paradigm, which learns the representation vectors of user and item

---

*Work done during an internship at Kuaishou Technology.

†Corresponding Author

39th Conference on Neural Information Processing Systems (NeurIPS 2025).

based on the assumption that "similar users exhibit similar behavior", where these vectors can be interpreted as latent topic distributions, analogous to those in Latent Dirichlet Allocation (LDA) [3] distribution. The cornerstone of this paradigm is the discovery and modeling of the item topics (*e.g.,* categories) and how to capture user preferences for these topics based on historical interactions.

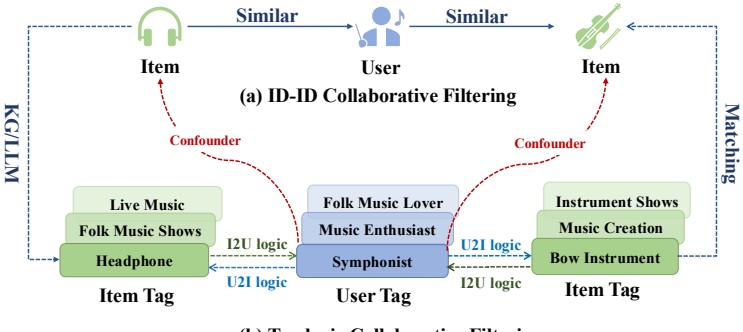

Figure 1: The toy example of the progress from traditional methods to tag-logic modeling.

Although effective, this paradigm neglects the modeling of user roles/characteristics and thus fails to capture the logical relationships between user roles and item types, which potentially restricts the expressiveness of the resulting recommendation model. On one hand, existing solutions may discover item-item correlation candidates that are hard to interpret by item types, but the user's role and personal characteristics may serve as the confounders, providing meaningful explanations for these correlations. A representative real-life example is the famous diaper-beer correlation [46], where a decent amount of human effort has been engaged to find that the "dads with newborns" are the logical explanation for the co-purchase behavior. On the other hand, interest-based modeling mostly relies on interest relations, while the user-item logic relations (*e.g.,* a certain type of user likes a certain type of item) can be far more interpretable and expressive. Consider the intuitive example in Figure 1, we observe a user who purchases a violin after consuming a headphone. While a statistical model may find the violin-headphone relation insignificant, the user may happen to be a symphonist, where both the symphonist-headphone edge and the symphonist-instrument edge are strong and general logical connections. As we will show in section 4.2, knowing the general logic in the real world and what role a user plays in real life may significantly improve the recommender's ability to engage in more accurate interest exploration [35].

**New Problems:** To mitigate the aforementioned limitations, we argue that the recommender system should complement the existing problem formulation with the following two tasks:

- The *user role identification task* that constantly identifies and models what roles the user plays in the real world (*e.g.,* "dads with newborns" and "symphonist"), different from pre-defined accessible user profile features like gender and age;
- The *behavioral logic modeling task* that models how user roles logically connect to the corresponding item topics. For this task, we further focus on two types of logic to align with the collaborative filtering paradigm as in Figure 1-b: 1) for a given user role, determine what types of items (also referred to as "topics") are suitable or interesting (*i.e.,* the *U2I logic*); And 2) for a given item topic, determine what kind of users would benefit from this content (*i.e.,* the *I2U logic*).

**Challenges:** 1) Different from item topic modeling [22, 4, 31], for practical and privacy concerns [18], user role identification is systematically challenging, for it is irresponsible, inefficient, and likely to be offensive to directly ask users to provide their social roles in many web services. Even if the users are willing to provide this information for mutual benefits, there is no guarantee that the provided features are accurate and comprehensive. 2) In terms of the logic modeling task, there have been some pioneering works that use user-generated hashtags or causal tag discovery methods with the help of human experts [47, 49, 50]. However, these methods do not accommodate the scale of industrial recommender systems. Furthermore, they heavily rely on high-quality but manually designed variables, which restricts the model's expressiveness in a large scale. Ideally, we would like to achieve an automatic modeling framework that can provide an immersive experience where the user roles and the task-specific logic patterns are modeled without bothering the users.

**Solution Framework:** Fortunately, Multi-modal Large Language Models (MLLMs) and Large Language Models (LLMs) have made significant breakthroughs [1, 64, 41, 54], demonstrating extensive world knowledge memorization abilities and advanced causal and logical reasoning capabilities [53, 65], which open the opportunities to reexamine the collaborative filtering framework's ability to model user roles and user behavioral logic. To this end, we propose a general solution framework *TagCF* that simultaneously solves the aforementioned tasks and improves the recommendation performance. Specifically, we first design a task-specific tag identification module utilizing an MLLM (*i.e.,* M3 [8]) to extract related user (role) tags and item (topic) tags for each given item, based on the semantic-rich multi-modal features. Then, starting from the identified set of user tags and item tags, we propose a virtual collaborative logic filtering module that uses another LLM (*i.e.,* Qwen2.5-7B [56]) to iteratively infer the U2I and I2U logic. To meet the scalability demand of the industrial environment, we propose several techniques, including cover set reduction and tag-logic model distillation. As we will discuss in Section 4.2.3, this logic graph presents general behavioral logic that can be transferred to other recommendation tasks.

Finally, the generated tag knowledge and the logic graph are integrated as enhancements for standard recommendation frameworks with three empirically effective designs: 1) For model architecture, we enhance item representations with a tag-based item encoder and propose a separate tag-based user encoding design to fulfill the user role identification task; 2) For learning augmentation, we further show that we can use a contrastive learning (CL) framework to integrate tag semantics into item and user representations; 3) During inference, we extend the recommendation model with a tag-logic inference score, which simultaneously boosts the recommendation accuracy and diversity.

**Empirical Support:** To verify the effectiveness of the tag extraction, the collaborative logic reasoning, and the recommendation enhancement framework, we conduct extensive experiments in an online A/B environment, an industrial offline dataset, and two public datasets. We also provide empirical findings on the different behaviors of user roles and item topics, ablation studies on model variants, and sensitivity analysis of hyperparameters.

## 2 Related Work

### 2.1 Collaborative Filtering

Collaborative filtering (CF), one of the most successful recommendation approaches, continues to attract interest in both academia and industry. Over time, CF has evolved from traditional methods [44, 33, 7, 5, 38] to advanced techniques incorporating sequences [25, 29, 48] and graph structures [51, 24]. Among the representative methods, matrix factorization (MF) techniques [5, 38] are effective in learning latent user and item representations. Sequential CF methods extend this by modeling the temporal order of user interactions with Recurrent Neural Networks [25] and Transformers [29, 48, 34]. Graph-based CF methods like NGCF [51] and LightGCN [24] have also gained attention in recent years. Besides, self-supervised learning approaches [58, 9, 60] have been explored to enhance CF by learning robust representations. However, these methods often ignore user roles and logical relationships between characteristics.

Meanwhile, some personality-aware filtering methods incorporate user traits through neighborhood filtering [30, 16, 15] or matrix factorization extensions [30, 16]. In the literature of psychology [21], the majority of the works used the Big-Five personality model to represent the user's personality, while the choice of the most suitable personality definition that satisfies the requirements of the recommendation application still needs further investigation. Recent works [32, 57, 45] have attempted to leverage LLMs for personalized recommendations and user interest interpretation. While progress has been made, existing approaches still overlook explicit modeling of user roles and their logical relationships. In this work, we aim to address these gaps by bridging topics and social roles via LLMs-enhanced logical recommendation within the CF framework.

### 2.2 LLM-based Recommendation

**LLM-enhanced Recommender.** Many current works have explored how to apply the LLM to generate auxiliary knowledge for enhancing traditional RS. LLMRG [52] fabricates prompts to construct chained graph reasoning from LLM to augment the recommendation model. LLMHG [12] first leverages LLMs to deduce Interest Angles (IAs) and categorize movies into multiple categories

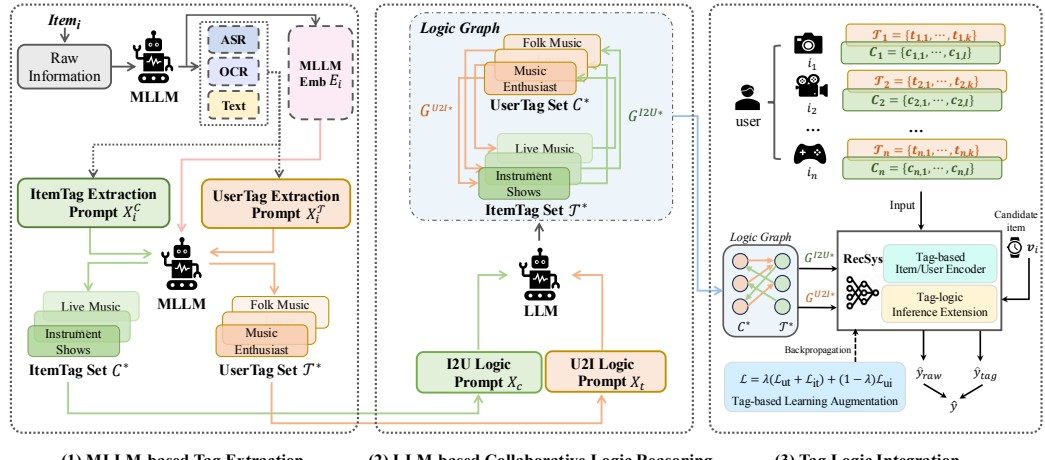

Figure 2: The main framework of the proposed *TagCF*.

within each IA to construct a multi-view hypergraph. SAGCN [36] uses a chain-based prompting strategy to extract semantic interactions from LLM for each review and introduces a semantic aspect-based graph convolution network to enhance the user and item representations by leveraging these semantic aspect-aware interactions. LLM-KERec [62] uses LLM to identify the complementary relationships of an item knowledge graph. Subsequently, they train an entity-entity-item weight decision model which is then used to inject knowledge into the ranking model by using the real exposure and click feedback of complementary items. Nevertheless, current methods of using LLM for data enhancement primarily focus on the meta-features, neglecting knowledge from the user side and the logic rationale between user-item interactions. This limitation hinders their ability to facilitate traditional recommenders to capture semantic and representative collaborative information.

**LLM as Recommender Itself.** Recently, LLMs have demonstrated remarkable performance across a wide range of recommendation tasks. P5 [20] and M6Rec [13] finetune LLM by modeling recommendation tasks as natural language processing tasks. ChatRec [17] employs LLMs as a recommender interface for conversational multi-round recommendations. TALLRec [6] designs a customized parameter-efficient tuning process for recommendation tasks on LLM with a LoRA architecture. HLLM [11] uses an item LLM to encode text features, feeding its embeddings to a user LLM for recommendations. Compared to LLM-enhanced recommender, this paradigm's computational cost (both for training and inference) is much higher and the industry-deployable solution is still an open question [14]. As we will discuss in section 3.3.1, this research direction focuses on the improvement of sequential models, which is complementary to our proposed knowledge extraction and augmentation framework.

## 3 The TagCF Framework

We present the task formulations of the standard top-N recommendation task, the user role (and item topic) identification task, and the behavioral logic reasoning task in Appendix A.1. The key notations in this paper are listed in Appendix A.2.

### 3.1 MLLM-based Item-wise Tag Extraction

For a given item $i \in \mathcal{I}$, we first take the original multi-modal information (*e.g.,* audio, image, and title of videos) and use a multi-modal LLM (MLLM), M3 [8], to generate a semantic item embedding $E_i$ and initial textual features. Then, we use the textual features to construct corresponding prompts $X_i^{\mathcal{T}}$ for user role tag extraction and $X_i^{\mathcal{C}}$ for item topic tag extraction (with prompt details in Appendix A.3). Given $E_i$ as auxiliary information, the given prompt will guide the generation of tags:

$$\mathcal{T}_i \sim \mathrm{M3}(X_i^{\mathcal{T}}, E_i); \mathcal{C}_i \sim \mathrm{M3}(X_i^{\mathcal{C}}, E_i), \tag{1}$$

where $\mathcal{T}_i$ and $\mathcal{C}_i$ are inferred user tags and item tags, and they are stored as static features of the given item $i$. In contrast, we assume that both the total user role tag set $\mathcal{T}$ and the total item topic tag set $\mathcal{C}$ change dynamically, so we apply update rules $\mathcal{T} \leftarrow \mathcal{T} \cup \mathcal{T}_i$ and $\mathcal{C} \leftarrow \mathcal{C} \cup \mathcal{C}_i$ on a daily basis.

**Unrestricted Tags and Cover Set Reduction:** A critical challenge of the application of Eq.(1) is the unrestricted open-world generation of tags that may gradually accumulate excessive tag sets, while the tag frequency could be extremely skewed (see Appendix C). To circumvent this problem, we propose a greedy and dynamic version of the min cover set finding algorithm (see Appendix A.4) to automatically find a small subset of expressive tags (*i.e.,* the cover set) that provide sufficient coverage of items and are mutually different in semantics. We denote the resulting cover sets of the two tag types as $\mathcal{T}^*$ and $\mathcal{C}^*$. In practice, we find that the cover set has some nice features in stability, generality, efficiency, and expressiveness (see Appendix C.1). All these features add up to the effectiveness of the extracted knowledge.

**Computational Bottleneck and Distillation:** In practice, another key challenge is the computational cost of the MLLMs, especially when there is a large number of newly uploaded items to process on a daily basis (*e.g.,* videos and news). As a countermeasure, we propose to apply Eq.(1) on a smaller subset (tens of thousands) of newly uploaded items, train efficient distilled models $P_\theta(t|i) : \mathcal{I} \times \mathcal{T}^* \rightarrow [0, 1]$ and $P_\theta(c|i) : \mathcal{I} \times \mathcal{C}^* \rightarrow [0, 1]$ based on the sampled data, then use $\theta$ to predict user/item tags for all items (in millions). We provide the algorithmic details of this procedure in Appendix A.4. In section 3.3, we show that $\theta$ may also participate in the recommendation model training, providing better alignment with user interactions.

## 3.2 LLM-based Collaborative Logic Filtering

With the daily update of $\mathcal{T}$ and $\mathcal{C}$, we use an LLM (*i.e.,* QWen2.5-7B [56]) to update and maintain the two graphs $\mathcal{G}^{\text{U2I}}$ and $\mathcal{G}^{\text{I2U}}$. Specifically, we iteratively select the tags that have not been included in the (source nodes of) logic graphs, construct the two logic reasoning prompts (in Appendix A.3), then obtain the I2U logic and U2I logic with:

$$\mathcal{T}_c \sim \text{LLM}(X_c); \mathcal{C}_t \sim \text{LLM}(X_t), \tag{2}$$

where $X_c$ and $X_t$ are input prompts for item tag $c \in \mathcal{C}$ and user tag $t \in \mathcal{T}$, $\mathcal{T}_c$ and $\mathcal{C}_t$ are generated tags, correspondingly. To keep the tag set update inclusive, we also update the tag sets with $\mathcal{T} \leftarrow \mathcal{T} \cup \mathcal{T}_c$ and $\mathcal{C} \leftarrow \mathcal{C} \cup \mathcal{C}_t$ on a daily basis. Both Eq.(1) and Eq.(2) use the pretrained model without finetuning in order to keep the intact world knowledge and reasoning ability, and we find the generation sufficiently accurate according to human expert justification (Appendix D.1).

**Distill Logic within Cover Sets:** As we have mentioned in section 3.1, we can achieve a stable and general inference using the cover sets $\mathcal{T}^*$ and $\mathcal{C}^*$. However, Eq.(2) does not guarantee a generation output within the cover sets. As a countermeasure, we learn distilled models $P_\varphi(c|t)$ and $P_\varphi(t|c)$ on the full tag sets with the LLM-inferred data generated by Eq.(2), then predict the logic connections between the cover sets $\mathcal{T}^*$ and $\mathcal{C}^*$, where the predicted scores are used to select top-$b$ target tags for each given input tag. We present the details of this process in Appendix A.5 and denote the resulting graph as $\mathcal{G}^{\text{U2I}*}$ and $\mathcal{G}^{\text{I2U}*}$. Additionally, as we will verify in Section 4.2.3, these graphs are transfer-friendly as they use tags of general concepts and each logic represents a real-world task-agnostic user behavioral logic, taking advantage of the LLM.

## 3.3 Tag-Logic Integration in Recommendation

Note that item tags and user tags emphasize different semantic aspects, one can implement two corresponding integration alternatives with symmetric design and we denote them as *TagCF-it* (that uses item tags to infer) and *TagCF-ut* (that uses user tags to infer). Without loss of generality, we introduce *TagCF-ut* with three effective augmentation methods in the following sections, and provide detailed specifications in Appendix A.6.

### 3.3.1 Tag-based Encoder

**Item Encoder:** For each item $i$, we first obtain user tags $\mathcal{T}_i$ and item tags $\mathcal{C}_i$ through $\theta$ (or Eq.(1)). Then, the embeddings of all extracted tags $\mathbf{T}_i = \{\mathbf{e}_t | t \in \mathcal{T}_i\}$ (or $\mathbf{C}_i = \{\mathbf{e}_c | c \in \mathcal{C}_i\}$ in *TagCF-it*) are aggregated through either Mean pooling or an Attention Mechanism [61] (the latter is adopted in practice), generating the tag-based item encoding $\mathbf{r}_i^{(t)} \in \mathbb{R}^d$ (or $\mathbf{r}_i^{(c)} \in \mathbb{R}^d$). These encodings provide

semantic information that may augment the standard ID-based item embedding $\mathbf{x}_i \in \mathbb{R}^d$. We provide the details of our attention operation in Appendix A.6.

**User Encoder:** For each user $u$, we first obtain the user's interaction history $\mathcal{H}_u$ as input. Then, we use two sequential models (*i.e.,* SASRec [29]), $\psi_x$ and $\psi_r$, that separately encode the ID-based item embeddings and the tag-based item embeddings for the history, and denote the resulting user encodings as $\mathbf{x}_u$ and $\mathbf{r}_u^{(t)}$ (*TagCF-it* generates $\mathbf{r}_u^{(c)}$ instead). Subsequently, we merge these two embeddings and obtain the enhanced user representation:

$$\phi_u = \text{MLP}_{\psi_u}(\mathbf{x}_u \oplus \mathbf{r}_u^{(t)}), \tag{3}$$

where $\oplus$ is the concatenation operation. Finally, we calculate the predicted score as:

$$\hat{y}_{\text{raw}}(u, i) = P(i|u) = \text{Sigmoid}(\phi_u^\top \mathbf{x}_i). \tag{4}$$

During training, each user history is associated with a set of interacted items $\mathcal{I}_u$ as positive targets, and we randomly sample a negative item $i^-$ for each $i^+ \in \mathcal{I}_u$. For each training sample $(u, i^+, i^-)$, the learning objective is defined as the combined binary cross-entropy loss:

$$\mathcal{L}_{\text{ui}}(u, i^+, i^-) = -w_{i^+} \log P(i^+|u) - \log(1 - P(i^-|u)), \tag{5}$$

where $w_{i^+}$ denotes the reward weight of the positive item. Intuitively, the combined user representation ensures the tag-aware encoding for both items and users, which improves the model expressiveness and recommendation accuracy.

### 3.3.2 Tag-based Learning Augmentation

In addition to the tag-aware encoders, we can also use the tag and logic information to provide augmented guidance through various training strategies. Similar to Eq.(5), we propose contrastive learning objectives on the tag space from both the user's perspective and the item's perspective:

$$
\begin{aligned}
\mathcal{L}_{\text{ut}}(u) &= -\sum_{t^+ \in \mathcal{T}_u^+} \log P(t^+|u) - \sum_{t^- \in \mathcal{T}_u^-} \log(1 - P(t^-|u)) \\
\mathcal{L}_{\text{it}}(i) &= -\sum_{t^+ \in \mathcal{T}_i^+} \log P(t^+|i) - \sum_{t^- \in \mathcal{T}_i^-} \log(1 - P(t^-|i)),
\end{aligned}
\tag{6}
$$

where $P(t|u) = \text{Sigmoid}(\phi_u^\top \mathbf{e}_t)$ estimates the probability of a user $u$ identified with a user role $t$, and $P(t|i) = \text{Sigmoid}(\mathbf{x}_i^\top \mathbf{e}_t)$ estimates the probability of an item $i$ being related to user role $t$. In practice, we can reuse $\theta$ in section 3.1 to realize the latter model $P(t|i)$. In the user level objective, $\mathcal{T}_u^+$ are user tags related to ground truth target items in $\mathcal{I}_u$, and $\mathcal{T}_u^-$ are tags related to sampled negative items. In the item level objective, $\mathcal{T}_i^+$ are user tags related to item $i$, and $\mathcal{T}_i^-$ are tags sampled from $\mathcal{T} \setminus \mathcal{T}_i^+$.

**Tag-Logic Exploration:** For the settings of $\mathcal{T}_u^+$, $\mathcal{T}_u^-$, and $\mathcal{T}_i^+$, we offer two alternatives that either use the original tag sets $\mathcal{T}_u(0) = \{t | t \in \arg_t \text{top-}k[P(t|u)]\}$ (or $\mathcal{T}_i(0) = \{t | t \in \arg_t \text{top-}k[P(t|i)]\}$ for $\mathcal{T}_i^+$) that address the recommendation utility (denoted as *TagCF-util*) or use the extended tag sets $\mathcal{T}_u(1)$ (or $\mathcal{T}_i(1)$) inferred by the logic graphs that address the interest exploration (denoted as *TagCF-expl*). For instance, we have a target item that has an initial tag $t = $ "Symphonist" which is logically related to the topic $c = $ "Music Theory" according to $\mathcal{G}^{\text{U2I}*}$. Then using $\mathcal{G}^{\text{I2U}*}$, we might explore and find that there exists a logic of "Music Theory" $\rightarrow$ "Teacher", where "Teacher" becomes the extended tag of the item. We provide a detailed description of the general procedure in Appendix A.6 and the confirmatory case study in Appendix C.3.

**Augmented Learning:** In summary, the augmented learning objective linearly combines the main objective with the two contrastive losses:

$$\mathcal{L}(u, i^+, i^-) = \mathcal{L}_{\text{ui}}(u, i^+, i^-) + \lambda\left(\frac{1}{|\mathcal{I}_u|}\mathcal{L}_{\text{ut}}(u) + \mathcal{L}_{\text{it}}(i^+)\right). \tag{7}$$

The resulting framework will align the item and user embedding space with the tag embedding space with $\lambda > 0$, which guides the model to match users and items according to the user tags.

### 3.3.3 Tag-logic Inference Extension

Despite the implicit tag modeling through learning augmentation, we also provide an explicit tag-logic inference strategy to further enhance recommendation performance and explainability. Specifically, we start from the user encoding $\phi_u$ from Eq.(3) and find the initial user tags of user $\mathcal{T}_u(0)$. Similar to the logical exploration process in section 3.3.2, we derive the extended tag set $\mathcal{T}_u(1)$ according to $\mathcal{G}^{\text{U2I}*}$ and $\mathcal{G}^{\text{I2U}*}$. Then, for each candidate item $i$, we can use the obtained user tags to calculate the tag-based matching score:

$$\hat{y}_{\text{tag}}(u,i,0) = \sum_{t \in \mathcal{T}_u(0)} P(t|u)P(t|i); \quad \hat{y}_{\text{tag}}(u,i,1) = \sum_{t \in \mathcal{T}_u(1) \backslash \mathcal{T}_u(0)} P(t|u)P(t|i), \tag{8}$$

where $P(t|i)$ and $P(t|u)$ are the same as those in Eq.(6). Finally, the overall score with explicit tag-logic inference extension becomes:

$$\hat{y}(u,i) = \hat{y}_{\text{raw}}(u,i) + \beta_0 \hat{y}_{\text{tag}}(u,i,0) + \beta_1 \hat{y}_{\text{tag}}(u,i,1). \tag{9}$$

where the Utility-based *TagCF-util* set $\beta_0 > 0, \beta_1 = 0$, and the Exploration-based *TagCF-expl* set $\beta_0 \geq 0, \beta_1 > 0$.

## 4 Experiments

### 4.1 Online A/B Test

#### 4.1.1 Workflow Specification

We conduct an online A/B test on a real-world industrial video recommendation platform to evaluate the effectiveness of *TagCF*-**ut**. The platform serves videos for over half a billion users daily, and the item pool contains tens of millions of videos. Figure 3 provides a detailed overview of the implementation of our online recommendation workflow. The tag extraction module, the collaborative logic reasoning module, and the training of all augmented models are offline procedures executed on a daily basis. In contrast, the inference part of the tag-logic integration module is deployed in the last ranking stage (which chooses top-6 scored items as recommendation from 120 candidates from the previous stage) for real-time recommendation requests, with preprocessed tag and logic information retrieved from the latest knowledge base. As we have described in Section 3.1 and Section 3.2, we use the cover set solution to achieve stable and efficient inference that fulfills the industrial demand.

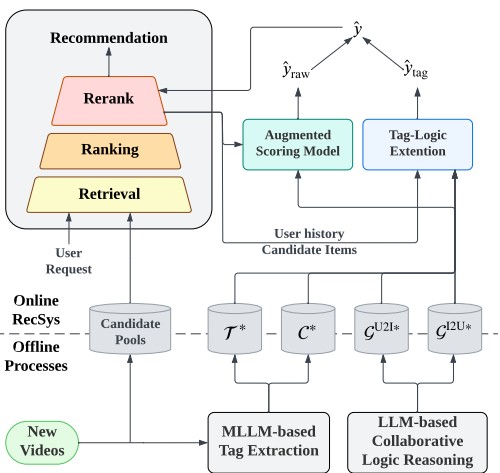

Figure 3: The deployment of *TagCF* in the online recommender system.

#### 4.1.2 Evaluation Protocol

For our online experiments, we randomly assign all users into 8 buckets, each accounting for relatively 1/8 of the total traffic, with each bucket consisting of tens of millions of users. We deploy *TagCF-util* and *TagCF-expl* in two different buckets and use two other buckets with the baseline model as comparisons. The baseline method (details omitted) in remaining buckets is a state-of-the-art ranking system that has been developed for four years from [29]. To ensure the reliability and validity of the experimental results, each method is subjected to an online testing phase of at least 14 days. To evaluate recommendation accuracy, we focus on the key interaction reward that combines positive user feedback (*e.g.,* effective play, like, follow, comment, collect, and forward). We also include the novelty-based diversity metric [2] that estimates the likelihood of recommending new video categories to a user, where the categories are predefined by human experts instead of the item tags in our framework to ensure fair comparison.

Table 1: Online performances of *TagCF* and ∗ denotes the results are statistically significant.

| Strategies | #Interaction | Diversity |
|---|---|---|
| *TagCF-util v.s.* baseline | **+0.946**% [*] | +0.001% |
| *TagCF-expl v.s.* baseline | +0.143% | **+0.102**%[*] |

#### 4.1.3 Effectiveness of Tag-Logic Augmentation in Practice

We summarize the results in Table 1 which shows that both *TagCF-util* and *TagCF-expl* outperform the baseline but exhibit different behaviors: *TagCF-util* significantly improves interaction metrics, which proves that the extracted tags can effectively represent the matching reasons and enhance recommendation accuracy. On the other hand, *TagCF-expl* significantly improves the diversity metric without losing recommendation accuracy, which proves that *TagCF-ut* can accurately explore user preferences through the logic graphs, mitigating the echo chamber effect. Moreover, we conducted an extended experiment for *TagCF-expl*, increasing the traffic to 2 buckets, and observed 40 days to validate the long-term effect. In addition to the improvement on the short-term diversity metric, we also observed a quantitatively and statistically significant boost of LT7 (a key metric that indicates long-term daily active users (DAU) and user retention benefits online in the next week) by 0.037%, proving the stable and consistent improvement on user satisfaction in the long run.

### 4.2 Offline Experiments

#### 4.2.1 Experimental Setup

**Datasets:** To further investigate the design choices of *TagCF*, we include two public datasets [39], Books and Movies, as well as an offline dataset from our real-world industrial video sharing platform (*i.e.,* Industry). More details about the datasets and preprocessing can be seen in Appendix B.2.

**Evaluation Protocol:** We include common ranking accuracy indicators such as NDCG@$N$ and MRR@$N$, as well as diversity metrics like ItemCoverage@$N$ and GiniIndex@$N$ (denoted as Cover@$N$ and Gini@$N$, respectively). In this paper, we observe $N \in \{10, 20\}$. For each experiment across all models, we run training and evaluation for five rounds with different random seeds and report the average performance.

**Baselines:** We include BPR [43] as the standard collaborative filtering method, and include several representative sequential models, namely GRU4Rec [25], Bert4Rec [48], SASRec [29], LRURec [59], Mamba4Rec [34]. We also compare with LLM-enhanced recommender approaches: RLM [42], SAID [27] and GENRE [37]. See more baseline details in Appendix B. We follow RecBole [63] as the implementation backbone and reproduce all baselines with hyper-parameters from either the original setting provided by authors or fine-tuning using validation.

#### 4.2.2 Effectiveness of Tag-Logic Integration

We present the overall experimental results in Table 2. Compared to BPR and sequential models, RLMRec and GENRE generally consistently improve the accuracy metric and, in most cases, improve the diversity, which are the best baseline methods. However, we can see that the improvement of these methods is not always statistically significant, especially in datasets with large scale (*e.g.,* Industry). Additionally, the BPR model outperforms other methods in the diversity metric by a large margin, but this comes with a severe sacrifice in recommendation accuracy. Excluding this exceptional model, our proposed *TagCF-it* and *TagCF-ut* consistently outperform all other baselines in accuracy and diversity metrics, providing extended verification for the expressiveness of the extracted tags and logic graphs, as well as the effectiveness of the tag-logic integration framework.

#### 4.2.3 Transferability Test

To validate that the extracted tags (*i.e.,* $\mathcal{T}^*$ and $\mathcal{C}^*$) and logic graphs (i.e.,$\mathcal{G}^{\text{U2I}*}$ and $\mathcal{G}^{\text{I2U}*}$) in our industrial solution encapsulate general knowledge, we conduct a cross-task transfer experiment. Specifically, we use the same tag extraction module in Eq.(1) to generate the data-specific tags for Books and Movies data. Then we use the semantic embedding [10] of these tags to find the closest tags in $\mathcal{T}^*$ and $\mathcal{C}^*$ so that the tag space is completely aligned. This means that the *TagCF* solutions

Table 2: Overall performance comparison on one offline Industry dataset and two public datasets. ↓: lower is better. The best performance is denoted in bold and the second is underlined (excluding the exceptional trade-off behavior of BPR in Books dataset). ∗: t-test with p-value < 0.005 and "Improv." denotes the improvements over the best baselines.

| Dataset | Method | NDCG@10 | NDCG@20 | MRR@10 | MRR@20 | Cover@10 | Cover@20 | Gini@10↓ | Gini@20↓ |
|---|---|---|---|---|---|---|---|---|---|
| Industry | MF-BPR | 0.0145 | 0.0215 | 0.0124 | 0.0147 | 0.1140 | 0.1682 | 0.9814 | 0.9720 |
| | GRU4Rec | 0.0177 | 0.0253 | 0.0118 | 0.0137 | 0.2364 | 0.3314 | 0.9656 | 0.9515 |
| | SASRec | 0.0182 | 0.0257 | 0.0121 | 0.0140 | 0.2704 | 0.3790 | 0.9617 | 0.9452 |
| | Bert4Rec | 0.0165 | 0.0232 | 0.0109 | 0.0125 | 0.2546 | 0.3577 | 0.9700 | 0.9561 |
| | LRURec | 0.0179 | 0.0262 | 0.0121 | 0.0143 | 0.3558 | 0.4763 | 0.9532 | 0.9372 |
| | Mamba4Rec | 0.0181 | 0.0253 | 0.0121 | 0.0142 | 0.3392 | 0.4489 | 0.9614 | 0.9452 |
| | RLMRec | 0.0180 | 0.0256 | 0.0122 | 0.0141 | 0.3312 | 0.4673 | 0.9575 | 0.9421 |
| | SAID | 0.0186 | 0.0264 | 0.0126 | 0.0145 | 0.3473 | 0.4723 | 0.9557 | 0.9398 |
| | GENRE | 0.0183 | 0.0262 | 0.0123 | 0.0142 | 0.3401 | 0.4602 | 0.9591 | 0.9417 |
| | _TagCF-it_ | 0.0198 | 0.0270 | **0.0134**∗ | **0.0155**∗ | 0.4013∗ | 0.5440∗ | 0.9316∗ | 0.9071∗ |
| | _TagCF-ut_ | **0.0201**∗ | **0.0276**∗ | 0.0132 | 0.0152 | 0.3832 | 0.5210 | 0.9370 | 0.9129 |
| | Improv. | +8.06% | +4.55% | +6.35% | +6.90% | +12.78% | +14.21% | +2.27% | +3.21% |
| Books | MF-BPR | 0.0633 | 0.0777 | 0.0481 | 0.0520 | **0.9636** | **0.9957** | **0.5511** | **0.5025** |
| | GRU4Rec | 0.1449 | 0.1644 | 0.1161 | 0.1214 | 0.6570 | 0.8116 | 0.7915 | 0.7558 |
| | SASRec | 0.1597 | 0.1800 | 0.1241 | 0.1297 | 0.7968 | 0.8999 | 0.7790 | 0.7536 |
| | Bert4Rec | 0.1515 | 0.1749 | 0.1008 | 0.1060 | 0.7326 | 0.8642 | 0.7940 | 0.7612 |
| | LRURec | 0.1549 | 0.1745 | 0.1198 | 0.1252 | 0.8236 | 0.9275 | 0.7529 | 0.7276 |
| | Mamba4Rec | 0.1641 | 0.1826 | 0.1330 | 0.1381 | 0.7970 | 0.9078 | 0.7767 | 0.7497 |
| | RLM | 0.1661 | 0.1872 | 0.1331 | 0.1389 | 0.7964 | 0.9071 | 0.7762 | 0.7507 |
| | SAID | 0.1705 | 0.1920 | 0.1373 | 0.1433 | 0.7992 | 0.9097 | 0.7695 | 0.7434 |
| | GENRE | 0.1674 | 0.1903 | 0.1332 | 0.1384 | 0.8213 | 0.9270 | 0.7749 | 0.7402 |
| | _TagCF-it_ | 0.1819 | 0.1998 | 0.1516 | 0.1565 | 0.8143 | 0.9311 | 0.7532 | 0.7247 |
| | _TagCF-ut_ | **0.1881**∗ | **0.2071**∗ | **0.1560**∗ | **0.1613**∗ | 0.8435∗ | 0.9399∗ | 0.7469∗ | 0.7194∗ |
| | Improv. | +10.3% | +7.86% | +13.60% | +12,56% | -12.40% | -5.60% | -26.21% | -30.15% |
| Movies | MF-BPR | 0.0574 | 0.0695 | 0.0432 | 0.0465 | 0.7692 | 0.8887 | 0.8170 | 0.7971 |
| | GRU4Rec | 0.1181 | 0.1275 | 0.1058 | 0.1083 | 0.6565 | 0.7977 | 0.8319 | 0.8060 |
| | SASRec | 0.1171 | 0.1271 | 0.1018 | 0.1045 | 0.8472 | 0.9183 | 0.7960 | 0.7867 |
| | Bert4Rec | 0.1118 | 0.1216 | 0.0994 | 0.1020 | 0.7925 | 0.9012 | 0.8331 | 0.8128 |
| | LRURec | 0.1201 | 0.1307 | 0.1051 | 0.1080 | 0.8786 | 0.9452 | 0.7746 | 0.7648 |
| | Mamba4Rec | 0.1193 | 0.1301 | 0.1047 | 0.1072 | 0.8098 | 0.8924 | 0.7905 | 0.7743 |
| | RLM | 0.1192 | 0.1304 | 0.1049 | 0.1076 | 0.8381 | 0.8912 | 0.7913 | 0.7738 |
| | SAID | 0.1210 | 0.1311 | 0.1057 | 0.1082 | 0.8397 | 0.8956 | 0.7975 | 0.7804 |
| | GENRE | 0.1206 | 0.1309 | 0.1053 | 0.1079 | 0.8563 | 0.9257 | 0.7715 | 0.7601 |
| | _TagCF-it_ | 0.1220 | 0.1310 | 0.1105 | 0.1128 | **0.8956**∗ | **0.9575**∗ | **0.7391**∗ | **0.7173**∗ |
| | _TagCF-ut_ | **0.1255**∗ | **0.1346**∗ | **0.1134**∗ | **0.1159**∗ | 0.8813 | 0.9540 | 0.7668 | 0.7490 |
| | Improv. | +3.72% | +2.67% | +7.28% | +7.12% | +4.59% | +1.30% | +4.20% | +5.63% |

for these two public datasets can skip the collaborative logic reasoning module and directly use $\mathcal{G}^{\text{U2I}*}$ and $\mathcal{G}^{\text{I2U}*}$ for tag exploration. The experimental results are presented in Table 2 and we observe that _TagCF_ variants consistently demonstrate superior recommendation accuracy and diversity in Books and Movies, proving its transferability to other tasks. Note that other LLM-based baselines also use the extracted tag-logic information to enhance recommendation in our experiments, which indicates that the knowledge is transferable to other methods as well.

### 4.2.4 Ablation Study

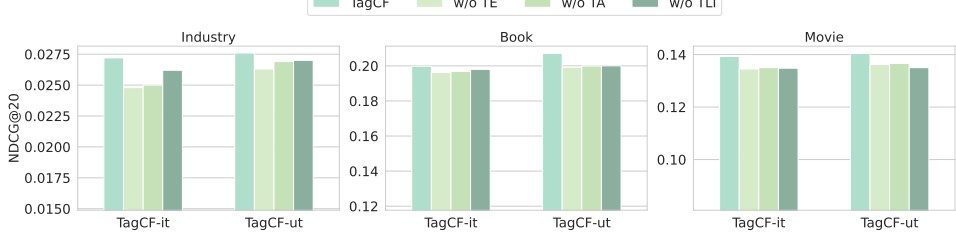

Figure 4: The ablation results of the three key methods of the tag-logic integration module.

- **Integration methods:** To evaluate the individual impact of the three main components in the integration framework (section 3.3), we compare the full _TagCF_ with three alternatives each disables one component in {tag-based encoder, tag-based learning augmentation, and tag-logic inference}, denoted as w/o TE, w/o TA, and w/o TLI, respectively. We show the results on the Industrial dataset in Figure 4, which verifies that all three components contribute to the recommendation accuracy.

The same conclusion applies to diversity metrics as well and the results are illustrated in Figure 8 of Appendix B.2. The performance degradation observed in each ablated variant underscores the complementary value of each module within the integrated framework.

- **Effect of $\beta_0$ and $\beta_1$:** We also analyses the impact of the inference scores of tags on recommendation performance. As shown in the figure 5, we varied the values of different weights $\beta_0$ and $\beta_1$ to analyze the effects of the original Utility-based tag score and Exploration-based tag score on the recommendation results.

- **Effect of $\lambda$:** We alter the $\lambda$ in Eq.(7) and present the results in Figure 6. We can see that there exists an optimal point in the middle, indicating the effectiveness of the learning augmentation.

- **Effect of $k$:** We conduct experiments with a different number of tags extracted for each item ($k \in \{20, 50, 100, 200, \text{full}\}$) and present the results in Figure 9 in Appendix B.2.



Figure 5: The model performance with different $\beta_0$ and $\beta_1$.

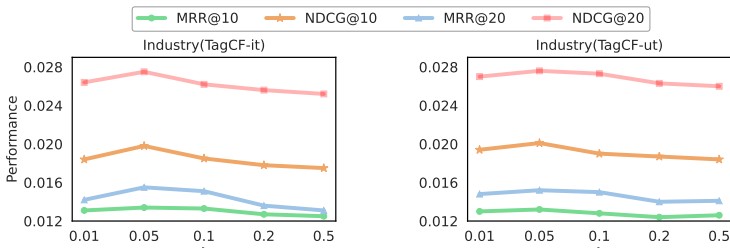

Figure 6: The model performance under different $\lambda$.

### 4.2.5 User Tag vs. Item Tag

In Table 2, we find that *TagCF-ut* tends to yield greater improvements in accuracy metrics, indicating that the user tag set is potentially more effective and stable in capturing preferences and personalities of users, solving the role identification task. This phenomenon might be related to the fact that user role tags are likely to be stable concepts with better expressiveness, which can be partially explained by the smaller cover set size compared with item tags shown in Table 6 of Appendix C.1. In contrast, item tags may have a shorter lifespan (*e.g.,* a special topic in recent news) and may frequently update even in the cover set. This may also explain the optimal diversity performance of *TagCF-it*, since the more fine-grained item tag set can contribute more diverse options during training and inference.

## 5 Conclusion

In this work, we emphasize the importance of the modeling of users' roles and the user-item behavior logic in the semantic tag space, and propose a new recommendation paradigm, *TagCF*, that can effectively extract item/user tags from items with MLLM, infer realistic behavioral logic of users with LLM, and enhance recommendation performance with the tag-logic knowledge. We provide technical details of our efficient and effective solution of *TagCF*, which has been successfully deployed in our industrial video-sharing platform. We also verify that the extracted knowledge of the logic graph is a general transferable asset to other recommendation tasks and LLM-based augmentation methods. Compared to the item tag set, the user role tags are empirically more stable and have more potential in improving recommendation accuracy, shedding light on an alternative design choice to the traditional item-tag-based methodology, posing new challenges to recommender systems.

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

# A  *TagCF* Specification

## A.1  Task Formulation

**Top-$N$ Recommendation Task** Define the set of users $\mathcal{U}$ and the set of items $\mathcal{I}$. The observed user-item interactions are represented as user histories $\mathcal{H}$, where each user's history $\mathcal{H}_u \in \mathcal{I}^{n_u}$ has length $n_u$. For top-$N$ recommendation, the objective is to learn a scoring function $P(i|u)$ that suggests top-$N$ items (from $\mathcal{I} \setminus \mathcal{H}_u$) that each user $u$ is highly likely to engage with, and the ground truth positive target item set is denoted as $\mathcal{I}_u$. Following the collaborative filtering paradigm, we assume a binary interaction label $y_{u,i}$, indicating the user $u$'s positive feedback on an item $i$, and we also allow an additional reward weight signal $w_{i+}$ for each positive item $i^+ \in \mathcal{I}_u$ to accommodate multi-behavior scenarios. In terms of the model input, we focus on the user history modeling which is related to the tag-based encoder in our solution. Yet, we remind readers that there may exist other context features that include but are not limited to user profile features, time features, device, and network features. How these features may integrate the tag-logic information is out of the scope of this paper, but worth further investigation.

**User/Item Tag Identification Task** As we have introduced in Section 3.1, $\mathcal{C}$ denotes the set of all possible item (topic) tags (e.g., headphone) and $\mathcal{T}$ denotes the set of all possible user (role) tags (e.g., symphonist). We assume that neither $\mathcal{C}$ nor $\mathcal{T}$ are known in advance, so we need an automatic inference framework to solve them. Recall that one of our focuses in this work is the user role identification task which formally finds a subset of user tags $\mathcal{T}_u \subset \mathcal{T}$ that describes a given user $u$. As discussed in Section 1, it is impractical to directly ask users to provide this information, but we can solve it by first figuring the user tags $\mathcal{T}_i \subset \mathcal{T}$ related to each item $i$, then learn a tag-based model to infer $\mathcal{T}_u$. This complements the conventional viewpoint that first associates the item topic tags $\mathcal{C}_i \subset \mathcal{C}$ to each item $i$, then predicts the item tag user profile $\mathcal{C}_u \subset \mathcal{C}$.

**Behavioral Logic Reasoning Task** With the discovered item tag set $\mathcal{C}$ and user tag set $\mathcal{T}$, we then solve the logical connections between them. Specifically, we aim to find a mapping $\mathcal{E}^{\text{U2I}} : \mathcal{T} \times \mathcal{C} \rightarrow [0, 1]$ that estimates the probability $P(c|t)$ of a certain U2I logic (i.e., a symphonist likes a violin), as well as a mapping $\mathcal{E}^{\text{I2U}} : \mathcal{C} \times \mathcal{T} \rightarrow [0, 1]$ that estimates the probability $P(t|c)$ of a certain I2U logic (i.e., a headphone is beneficial to a symphonist). These two mapping functions semantically define the edges of a directed logic graph between $\mathcal{C}$ and $\mathcal{T}$, and we denote the corresponding sub-graphs as $\mathcal{G}^{\text{U2I}} = (\mathcal{V}, \mathcal{E}^{\text{U2I}})$ and $\mathcal{G}^{\text{I2U}} = (\mathcal{V}, \mathcal{E}^{\text{I2U}})$, where $\mathcal{V} = \mathcal{T} \bigcup \mathcal{C}$. As a practical assumption, we do NOT assume that the two tag sets are mutually exclusive, *i.e.,* $\mathcal{T} \bigcap \mathcal{C} \neq \emptyset$, since a user role might be considered as a topic as well (e.g. a video about a symphonist). Same as the tag identification task, there is no ground truth label in this task, and we will take advantage of the generation and reasoning ability of LLMs to approximate the actual behavior logic.

## A.2  Notations and Terminologies

We summarize the key notations used in this paper in Table 3. Additionally, we find that "topic" and "interest" are two semantically confusing terms that both express the item type tags. In our paper, we refer to "topic" as the item type in the view of an item (*i.e., $P(c|i)$*) and "interest" as the item type in the view of a user (*i.e., $P(c|u)$*).

## A.3  Prompt Designs

We provide the prompt design details for the tag extraction task in Section 3.1 and present examples of the MLLM response in Table 4. The input textual features of [Title]/[ASR]/[OCR] are preprocessed text from the MLLM that describes the contents of the item, and we remind readers that this design might be task specific (*e.g.,* Books datasets only uses [Title]).

Then, we provide the prompt design details for the collaborative logic filtering module in Section 3.2 and present examples of the LLM's output in Figure 7. In practice, we find that including an intuitive example with input and output significantly improves the interpretability and the recognition rate of tags during post-processing.

## A.4  Tag Extraction Algorithm

In section 3.1, we introduce the process of cover set reduction which aims to find a small subset of the full tag set that can cover a sufficient number of items while ensuring the semantic differences between tags. We present the algorithm in Alg.1 and the process runs on a daily basis. The process iteratively includes a new tag into the cover set, and each newly included tag maximizes the coverage on the uncovered items (line 6) until no less than $\tau = 99\%$ of the items has at least one tag included in the cover set. Tags that have not been recalled by any item in the last $\mathbb{D}$ days will be removed (line 11), indicating an out-of-date tag. In practice, we observe that the cover set converges (less than 10 tag removed or added per day) after 30 days of updates. The statistics of the resulting tag sets and their cover sets are summarized in Table 6 in Appendix C.1.

As we have discussed in Section 3.1, we assume a computational bottleneck during inference of Eq.(1), which indicates that the system can only support the MLLM inference on a subset $\mathcal{I}' \subset \mathcal{I}$ (around 500,000 items per

Table 3: Key Notations

| Symbol | Description |
| --- | --- |
| $\mathcal{U}, \mathcal{I}$ | set of users and items |
| $u, i$ | specific user and item |
| $\mathcal{H}_u$ | interaction history of user $u$ |
| $\mathcal{I}_u$ | positive target items of user $u$ |
| $\mathcal{T}, \mathcal{C}$ | set of user role tags and item topic tags |
| $\mathcal{T}^*, \mathcal{C}^*$ | the extracted cover sets in section 3.1 |
| $\mathcal{T}_u, \mathcal{T}_i$ | user role tags inferred for user $u$ and item $i$, correspondingly |
| $\mathcal{C}_u, \mathcal{C}_i$ | item topic tags inferred for user $u$ and item $i$, correspondingly |
| $\mathcal{T}_u(0), \mathcal{C}_u(0)$ | the initial inferred tag sets from user |
| $\mathcal{G}^{\text{U2I}}, \mathcal{G}^{\text{I2U}}$ | the logic graphs extracted on the full tag sets |
| $\mathcal{G}^{\text{U2I}*}, \mathcal{G}^{\text{I2U}*}$ | the logic graphs extracted on the cover sets |
| $\mathcal{E}^{\text{U2I}}, \mathcal{E}^{\text{I2U}}$ | the edge mappings for logic graphs |
| $P_\theta(t\|i), P_\theta(c\|i)$ | The distilled model for tag extraction in Section 3.1 |
| $P_\varphi(c\|t), P_\varphi(t\|c)$ | the distilled model for U2I and I2U logic prediction in Section 3.2 |
| $P(i\|u), P(i\|\mathcal{H}_u)$ | the inferred likelihood of engagement for the user-item pair |
| $P(t\|u), P(c\|u)$ | user role and item interest prediction of user $u$ |
| $P(t\|i), P(c\|i)$ | user role and item topic prediction of item $i$ |
| $\mathbf{e}_t, \mathbf{e}_c$ | tag embedding of a specific user role and item topic |
| $\mathbf{T}_i, \mathbf{C}_i$ | the sets of tag embeddings related to an item |
| $\mathbf{r}_i^{(t)}, \mathbf{r}_i^{(c)}$ | item embedding inferred by tag-based encoder |
| $\mathbf{x}_i$ | ID-based item embedding |
| $\mathbf{r}_u, \mathbf{x}_u$ | user embedding inferred by tag embedding and ID embedding sequence |
| $\phi_u$ | final user embedding from the user encoder |

Table 4: The prompt templates for item tag identification and user role tag identification

| Item Tag Extraction Prompt Template | MLLM Response Example |
| --- | --- |
| This is the video's [Title] / [ASR] / [OCR] information. To make the video interesting for users, please extract 8-10 independent and detailed interest tags based on the multimodal contents. | [Pet Videos; Family Warmth; Song Cover Challenge; Pet Companionship; Music Production; Newborn Puppy; Cute Style; Daily Life] |
| **User Tag Extraction Prompt Template** | **MLLM Response Example** |
| This is the video's [Title] / [ASR] / [OCR] information. Identify 8-10 distinct target audience segments that would find this video appealing, such as "xx family," "xx professionals," or "xx enthusiasts." | [Fashion Enthusiast; Beauty Influencer; Fashion Designer; Personal Image Consultant; Hairstylist; Fashion Critic; Internet Celebrity; Fashion Photographer] |

day), and we learn a distilled model $\theta$ to solve the tag extraction problem for the remaining items in $\mathcal{I} \setminus \mathcal{I}'$. Without textual generation, we find that the multi-modal embedding model [10] is sufficiently efficient to infer all newly uploaded items each day, and it is reasonable to believe that the output $E_i$ contains sufficient information of the item to accurately infer the corresponding tags. Thus, we adopt $P_\theta(t|i) = P_\theta(t|E_i)$ and $P_\theta(c|i) = P_\theta(c|E_i)$.

## A.5 Logic Reasoning Process

In Section 3.2, we have introduced the collaborative logic filtering task and proposed to infer the logic graph in the cover set with distilled models. Specifically, when inferring logically related tags for a given source tag using Eq.(2), the output tags may or may not appear in the cover set due to the unrestricted open world generation. In practice, we find that the generated tags rarely match those tags in the cover set, but it is likely to find semantically close alternatives. Thus, we train distilled models $P_\varphi(c|t) : \mathcal{T} \times \mathcal{C} \to [0, 1]$ and $P_\varphi(t|c) : \mathcal{C} \times \mathcal{T} \to [0, 1]$ based on the offline data generated by Eq.(2) each day. The models take the semantic embedding of tags as input and output the likelihood of logical connection between the two (full) sets $\mathcal{C}$ and $\mathcal{T}$. After the daily training, we use $P_\varphi(c|t)$ to predict scores of $c \in \mathcal{C}^*$ with the given source tag $t \in \mathcal{T}^*$, and use $P_\varphi(t|c)$ to predict scores of $t \in \mathcal{T}^*$ with the given source tag $c \in \mathcal{C}^*$. Empirically, we observe that the top-50 predicted tags are semantically accurate logical connections in most cases, and the top-20 predicted tags are sufficiently diverse. Thus, we adopt the top-20 connections as edges in $\mathcal{G}^{\text{U2I}*}$ and $\mathcal{G}^{\text{I2U}*}$.

**(a) U2I Reasoning Instruction and LLM Output**     **(b) I2U Reasoning Instruction and LLM Output**

Figure 7: The instructions for collaborative logic reasoning.

---

**Algorithm 1:** Dynamic Cover Set Reduction Algorithm

---

**Input:** Most up-to-date cover set $\mathcal{S} = \mathcal{T}^*$ (or $S = \mathcal{C}^*$ in *TagCF-it*) (set to $\emptyset$ if not exist); Newly inferred item-tag mapping $M$ within cover set $\mathcal{S}$; The $\mathbb{D}$ day tag-item history $H$ of tags in cover set $\mathcal{S}$

1:   $\mathcal{I}_{\text{covered}} \leftarrow$ find all items in $M$ that have been covered by $\mathcal{S}$ and report tag recall rate;
2:   $\mathcal{S}_{\text{select}} \leftarrow$ find all tags in $\mathcal{S}$ that have covered items in $M$;
3:   $\mathcal{I}_{\text{new}} \leftarrow$ find all items appeared in $M$;
4:   $\mathcal{S}_{\text{new}} \leftarrow$ find all tags appeared in $M$;
5:   **while** $|\mathcal{I}_{\text{covered}}|/|\mathcal{I}_{\text{new}}| < \tau$ **do**
6:     Find the tag $t \in \mathcal{S}_{\text{new}} \setminus \mathcal{S}_{\text{select}}$ that covers the most number of items in $\mathcal{I}_{\text{new}} \setminus \mathcal{I}_{\text{covered}}$;
7:                   ▷ This ensures semantic differences between selected tags in $\mathcal{S}$
8:     $\mathcal{S}_{\text{select}} \leftarrow \mathcal{S}_{\text{select}} \cap \{t\}$ and update $\mathcal{I}_{\text{covered}}$ with newly covered items;
9:   **end while**
10: Append a new day history to $H$ with data in $M$ and remove the history in the oldest date;
11: Remove tags in $\mathcal{S}_{\text{select}}$ that have no records in $H$;
12: Store updated cover set $\mathcal{T}^* \leftarrow \mathcal{S}_{\text{select}}$ (or $\mathcal{C}^* \leftarrow \mathcal{S}$) and the updated tag-item history $H$.

---

Different from cover sets that quickly converge in size, the full tag sets continuously expand themselves and the same happens to corresponding logic graphs. Although one can assume that the possible tags in the open world are limited and expect the graphs to converge eventually, we notice that the 30-day inference already generates a graph too large to be directly used under the latency requirement. Additionally, the majority of the tags in the full set as well as their corresponding logic connections are usually fine-grained with very strong interpretability, but only cover a small set of items or user behaviors with undesirable generalizability. In general, we believe that the cover set tag-logic better suits the statistical models in the recommender systems, while the full set tag-logic is a better choice for detailed explanation.

### A.6 Augmentation Model Specification

**Details of Tag-based Encoders:** Define the user tag embedding sequence as $\mathbf{T}_i = [\mathbf{e}_{t_1}, \mathbf{e}_{t_2}, \ldots, \mathbf{e}_{t_k}] \in \mathbb{R}^{d \times k}$ where each tag is associated with a learnable $d$-dimensional vector. Similarly, define the item tag embedding sequence as $\mathbf{C}_i = [\mathbf{e}_{c_1}, \mathbf{e}_{c_2}, \ldots, \mathbf{e}_{c_k}] \in \mathbb{R}^{d \times k}$. Then, we calculate the tag-based item encoding $\mathbf{r}_i^{(t)} \in \mathbb{R}^d$ and $\mathbf{r}_i^{(c)} \in \mathbb{R}^d$ by fusing the tag embeddings using *item encoders*:

$$\mathbf{r}_i^{(t)} = f(\mathbf{T}_i), \mathbf{r}_i^{(c)} = g(\mathbf{C}_i). \tag{10}$$

The fusion functions $f$ and $g$ can be accomplished through methods such as the Mean Pooling or Attention Mechanism [61]. We adopt the latter attention mechanism in practice to model the different importance of each tag and the mutual influences between tags. Specifically, taking user role tags (in *TagCF-ut*) as an example, the

adopted Attention operation is formulated as:

$$\mathbf{r}_i^{(t)} = \boldsymbol{\alpha}_i \mathbf{T}_i, \quad \boldsymbol{\alpha}_i = \text{softmax}\left(\mathbf{W}\mathbf{T}_i + \mathbf{b}\right) \tag{11}$$

where $\mathbf{W} \in \mathbb{R}^{d \times d}$ and $\mathbf{b} \in \mathbb{R}^d$ are learnable parameter weights. $\boldsymbol{\alpha}_i$ is the tag attention score. This formulation enables the model to prioritize informative tags while suppressing noise, enhancing the discriminative power of the resulting tag-based item representation $\mathbf{r}_i^{(t)}$.

Then for a given user and the corresponding history $\mathcal{H}_u = \{i_1, \cdots, i_n\}$, we first obtain the standard ID-based item embedding sequence $\mathbf{X}_u = [\mathbf{x}_{i_1}, \cdots, \mathbf{x}_{i_n}] \in \mathbb{R}^{d \times n}$ and the tag-based item embedding sequence $\mathbf{R}_u^{(t)} = [\mathbf{r}_{i_1}^{(t)}, \cdots, \mathbf{r}_{i_n}^{(t)}] \in \mathbb{R}^{d \times n}$ each obtained from the tag-based item encoder, *i.e.,* Eq.(10). We then include two SASRec-style [29] *user encoder* networks with identical architecture which first obtain separate hidden embeddings of the user:

$$\begin{aligned} \mathbf{p}_u &= \text{ItemSASRec}(\mathbf{X}_u), \\ \mathbf{r}_u^{(t)} &= \text{TagSASRec}(\mathbf{R}_u^{(t)}), \end{aligned} \tag{12}$$

where $\mathbf{p}_u \in \mathbb{R}^d$ and $\mathbf{r}_u^{(t)} \in \mathbb{R}^d$ (*TagCF-it* outputs $\mathbf{r}_u^{(c)} \in \mathbb{R}^d$ instead). Note that one can also try other user encoding schemes such as item embedding concatenation or addition followed by a single encoder, but empirically, we find that the separate encoder networks yield the best results.

**Tag-Logic Exploration in Learning and Inference Augmentation:** Without loss of generality, we explain the exploration strategy on user role tags in *TagCF-ut* as the extended description of Section 3.3.2 and 3.3.3, and the solution in *TagCF-it* is symmetric. We start by considering the *initial tag set* $\mathcal{T}(0)$ that focuses on improving utility (*i.e., TagCF-util*), where $\mathcal{T}(0)$ may represent $\mathcal{T}_i(0)$ for a given item (inferred from $P(t|i)$) or $\mathcal{T}_u(0)$ for a given user (inferred from $P(t|u)$). Then we can use the U2I logic graph to find logically related item topic tags as $\mathcal{C}(1) = \{c | \exists t \in \mathcal{T}(0), \text{s.t.} (t,c) \in \mathcal{E}^{\text{U2I*}}\}$. Note that we use the distilled model $\varphi$ to generate graphs, and the corresponding scores could be used as weights of the edges. In this case, it can also use a soft method that selects the tags with aggregated weights as $\mathcal{C}(1) = \{c | w_c > \delta\}$, where $w_c = \sum_{t \in \mathcal{T}(0), (t,c) \in \mathcal{E}^{\text{U2I*}}} P(c|t)$. Finally, we can obtain the final *exploration tag set* $\mathcal{T}(1)$ by applying I2U logic on $\mathcal{C}(1)$ as $\mathcal{T}(1) = \{t | \exists c \in \mathcal{C}(1), \text{s.t.} (c,t) \in \mathcal{E}^{\text{I2U*}}\}$. Again, the corresponding soft method gets $\mathcal{T}(1) = \{t | w_t > \delta\}$, where $w_t = \sum_{c \in \mathcal{C}(1), (c,t) \in \mathcal{E}^{\text{I2U*}}} w_c$. To better align the scale of weights in $\mathcal{T}(0)$ and $\mathcal{T}(1)$, we normalize the weights so that they sum up to one. For better illustration of this process, we further provide case studies of the difference between $T(0)$ and $T(1)$ in Appendix C.3.

Note that with an average branch factor $b$, we would observe $|\mathcal{T}(1)| = O(b^2 k)$, which is several magnitudes larger than the initial set, so we truncate the top-$k$ tags in $\mathcal{T}(1)$ according to the frequency or weights to reduce noise, resulting in $|\mathcal{T}(1)| = |\mathcal{T}(0)|$. In practice, we can achieve fast computation of these processes by representing the graphs as sparse adjacency matrices and engaging multiplication with parallel computing.

# B    Experimental Settings

## B.1    Online Experiments

**Implementation Details.** We conduct an online A/B test on a real-world industrial video recommendation platform to evaluate the effectiveness of our method. The platform serves videos for over half a billion users daily, and the item pool contains tens of millions of videos. The number of candidates for each request in this stage is 120 and the videos with top-6 scores are recommended to users. To ensure that tag encompasses over 90% of user video views, we process 3 million videos daily by tag extraction module deployed on a cluster of 50 NVIDIA 4090 GPUs.

**Evaluation Protocol.** For our online experiments, we randomly assign all users into 8 buckets, each accounting for relatively 1/8 of the total traffic, with each bucket consisting of tens of millions of users. We deploy *TagCF-util* and *TagCF-expl* in two distinct buckets, while reserving two additional buckets for the baseline model comparison. The remaining buckets employ a state-of-the-art ranking system (details omitted for brevity) that has been iteratively optimized over four years [29]. To ensure statistical reliability, each experimental condition undergoes a minimum 14-day online testing phase. To evaluate recommendation accuracy, we focus on the key interaction reward that combines positive user feedback (*e.g.,* effective play, like, follow, comment, collect, and forward). We also include the novelty-based diversity metric [2] that estimates the likelihood of recommending new video categories to a user, where the categories are predefined by human experts instead of the item tags in our framework to ensure fair comparison.

## B.2    Offline Experiments

**Datasets.** We include two public datasets [39], Books and Movies, as well as an offline dataset from a real-world industrial video sharing platform (i.e., Industry). For public datasets, we utilize product descriptions as textual

Table 5: The statistics of the datasets.

| Dataset | #Users | #Items | #Interactions | #Sparsity |
|---------|--------|--------|---------------|-----------|
| Books | 9,209 | 8,299 | 935,958 | 98.77% |
| Movies | 39,832 | 24,050 | 1,103,918 | 99.88% |
| Industrial | 89,417 | 10,396 | 3,292,898 | 99.64% |

features and filter out products without descriptions. We convert the ratings of 3 or larger as positive interactions. For the Industrial dataset, we first select around 10k photos and obtain audio, visual, and textual features of each video. Then we take the user interactions on these photos in one day as the training set and those in the next day as the test set (excluding unseen users). To ensure the quality of the dataset, we follow the common practice [48, 23, 29] and keep users with at least ten interactions through n-core filtering. The statistics of the processed datasets are summarized in Table 5.

**Evaluation Protocol.** We include common ranking accuracy indicators such as NDCG@$N$ and MRR@$N$, as well as diversity metrics like ItemCoverage@$K$ and GiniIndex@$N$ (denoted as Cover@$N$ and Gini@$N$, respectively). In this paper, we observe $N \in \{10, 20\}$. For each experiment across all models, we run training and evaluation for five rounds with different random seeds and report the average performance.

**Baselines.** We include BPR [43] as the standard collaborative filtering method, and include several representative sequential models, namely GRU4Rec [25], Bert4Rec [48], SASRec [29], LRURec [59], Mamba4Rec [34]. We also compare with competitve LLM-enhanced recommendation methods: RLM [42] integrates representation learning with LLMs and aligns the semantic space of LLMs with the representation space of collaborative relational signals. SAID [27] utilizes LLMs to explicitly learn semantically aligned item ID embeddings based on texts for practical recommendations. GENRE [37] employs prompting techniques to enrich the training recommendation data at the token level to boost content-based recommendation.

**Implementation Details.** All experiments for recommender systems in this paper are conducted on the Tesla V100 GPUs. In the experiment, the MLLM used for item-wise tag extraction is M3 [8] and the LLM used for tag logic inference is Qwen2.5-7B-Instruct [56]. The LLM semantic embedding models used for the LLM-based baselines is text-embedding-3-small [40] from OpenAI. For *TagCF* training, we use the Adam optimizer with a learning rate of 1e-3 and weight decay of 1e-5. We follow RecBole [63] as the implementation backbone and reproduce all baselines with hyper-parameters from either the original setting provided by authors or fine-tuning using validation. For our user encoder and tag-based item encoder, we use two layers SASRec with hidden size of 256 and head size of 2.

**Ablations.** To assess the individual contributions of the three key components in our integration framework, we conduct an ablation study comparing the complete *TagCF* system with three variants, each excluding one component: tag-based encoder, tag-based learning augmentation, tag-logic inference, denoted as w/o TE, w/o TA, and w/o TLI, respectively. As demonstrated in Figure 8, the experimental results confirm that all three components significantly enhance both recommendation accuracy and diversity metrics. The performance degradation observed in each ablated variant underscores the complementary value of each module within the integrated framework.

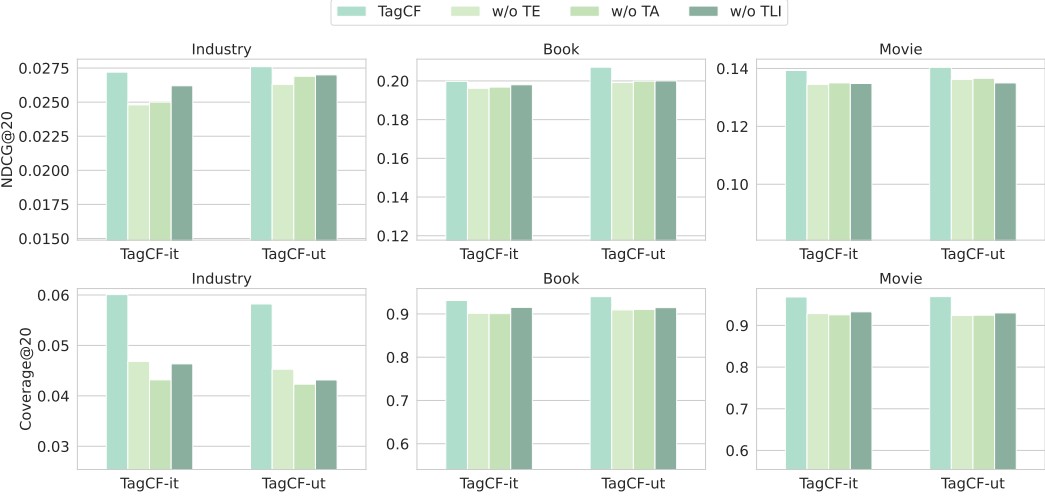

Figure 8: The ablation results of the three key methods of the tag-logic integration module.

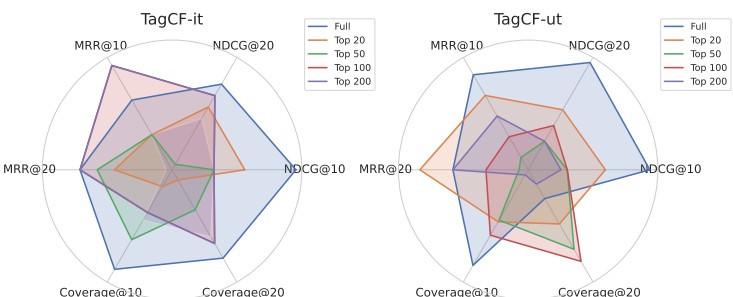

Figure 9: The impact of the number of top-$k$ during inference.

We also conduct experiments with a different number of tags extracted for each item ($k \in \{20, 50, 100, 200, \text{full}\}$) and present the results in Figure 9. Though it might be impractical for industrial solutions, we find that the full tag set achieves the best results, where *TagCF-it* tends to focus on diversity metrics and *TagCF-ut* addresses the accuracy metrics.

## C   Observations and Analysis

### C.1   Statistics of Tag Sets

Table 6 summarizes the statistics of the full tag set of $\mathcal{T}$ and $\mathcal{C}$, as well as the reduced cover set $\mathcal{T}^*$ and $\mathcal{C}^*$. Based on the statistics provided in Table 6, we find that item tags generally have a shorter lifespan compared to user tags. While the full tag sets for both users and items continuously expand without removal, the daily expansions reveal key distinctions. Item tags exhibit a significantly higher daily expansion, indicating more frequent updates. In contrast, user tags have a much smaller daily expansion and have nearly converged in the cover set. This smaller set size with less frequent expansion suggests that user tags are more stable and have a longer lifespan, whereas the high update frequency of item tags points to a shorter lifespan. In practice, we also find that tags follow an extremely skewed frequency distribution (Figure 11), indicating that not all tags are identically useful and expressive. While a few general tags may be retrieved by a large number of items, there also exists a large number of precise but unique tags that cannot cover a sufficient number of items. As illustrated in section 3.1, this motivates our design of the cover set reduction module. In practice, the open full set size tends to expand at a considerable rate even after the 30-day observation period, while the reduced cover set quickly converges in the first few days.

Table 6: Tag set statistics in our industrial platform

| type | full size | daily expansion | cover set size | cover set daily expansion |
|------|-----------|-----------------|----------------|---------------------------|
| user tag | 2,976,845 | 200-300K | 7,633 | converged |
| item tag | 50,208,782 | 3.5-4.0 million | 20,956 | hundreds |

**Tag Case Study:** Figure 10 presents a case study comparing the original video content with its corresponding generated tags. The figure demonstrates that both the user tags and item tags produced by the MLLMs are highly expressive and of superior quality, effectively capturing the video's key attributes.

To validate the reasonableness of the tag set distribution extracted by the MLLMs, we analyze the frequency distribution of tags, as illustrated in Figure 11.

**Tag Frequency Distribution (Left Plot):** The left plot shows the tag frequency distribution, where the $x$-axis represents individual tags ordered by their IDs, and the $y$-axis corresponds to the log-scaled frequency of occurrence. From the tag frequency plot, we observe the distribution follows a pattern consistent with real-world tag systems (e.g., a power-law distribution), as both UserTag and ItemTag curves exhibit a steep decline in frequency as Tag ID increases. This indicates that: A small subset of tags dominates (e.g., common tags like "elegant" or "cheap"). Long-tail tags (high Tag ID) are rare but exist, indicating diversity in generated tags.

**Tags per Item Distribution (Right Plot):** The right plot displays the distribution of tags per item, with the $x$-axis showing the number of tags per item and the $y$-axis representing the log-scaled frequency of items associated with each tag count. The peak observed at 1.0-1.5 (log scale) suggests that most items are assigned 3–5 tags (since $10^{1.0} \approx 3, 10^{1.5} \approx 5$). This balanced tagging behavior, neither overly sparse nor excessive, enhances the usability of the generated tags for downstream recommendation tasks.

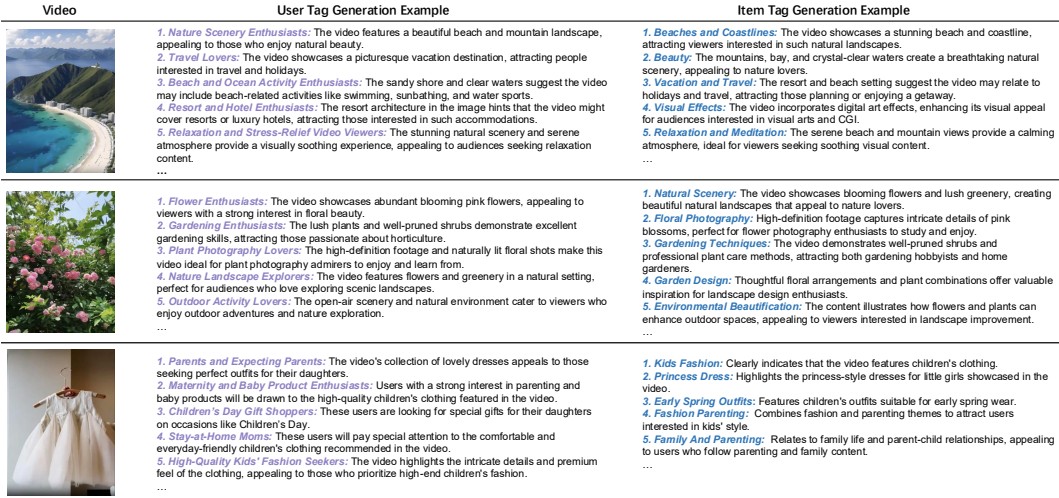

Figure 10: The example of tags generated by the MLLMs.

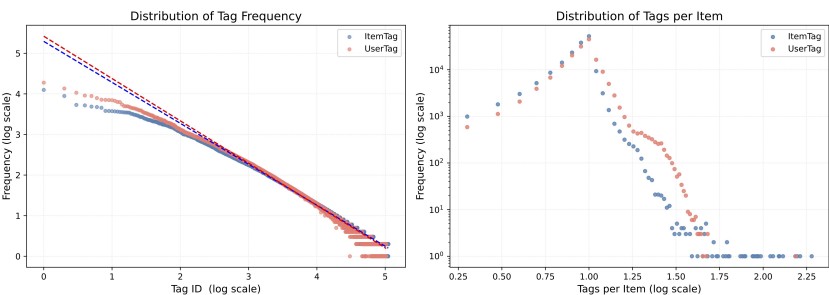

Figure 11: Item tag and user tag frequency distribution.

## C.2 Statistics of Logic Graph

We also investigate the quality of the logic graph by analyzing the edge degree of the U2I and I2U logic graph in Figure 12. The left is a scatter plot with marginal distributions of the U2I graph. Since the U2I graph represents a directed graph of user tags to item tags conversion relationships, the x-axis indicates the out-degree of use tags, and the y-axis indicates the in-degree of item tags. The right is a scatter plot with marginal distributions of the I2U graph, with the x-axis indicates the out-degree of item tags and the y-axis indicates the in-degree of user tags. We observe that in the U2I graph, the out-degree distribution of user tags is highly dispersed, indicating that user-generated tags reflect personalized social roles rather than conforming to homogeneous labeling patterns. This suggests that the divergent logic of the U2I graph can cover a broader range of item tags, mitigating the "clustering effect" and thereby breaking through information filter bubbles to enhance the diversity of recommendation results.

## C.3 Case Study of Tag-Logic Exploration

Intuitively, the initial tags $\mathcal{T}_i^{(0)}$ represent the most obvious type of users that the item would match, while $\mathcal{T}_i^{(1)}$ diverges from the initial user roles which tend to explore outside the echo chamber [19] in the recommendation process. We provide a real case example in Figure 13 to illustrate the differences. We consider both sets as effective tags of the corresponding item and we define positive/negative tags for the positive/negative items as:

$$\mathcal{T}_{i+} = \mathcal{T}_{i+}^{(0)} \cup \mathcal{T}_{i+}^{(1)}, \mathcal{T}_{i-} = \mathcal{T}_{i-}^{(0)} \cup \mathcal{T}_{i-}^{(1)}, \tag{13}$$

where the weights of the same tag are summed and normalized.

We further present a case study in Figure 13 that illustrates the transformation from users' original tags to exploratory tags during the inference process. Specifically, we compare: (1) the original user tags predicted by the model for the target user, and (2) the logically explored user tags from the initial user tags. The results demonstrate that the purple-highlighted tags in the exploration tag set (Wedding Preparers, Parent-child Activity Participants and Skiing Enthusiast) successfully break through the information cocoon of the original tag

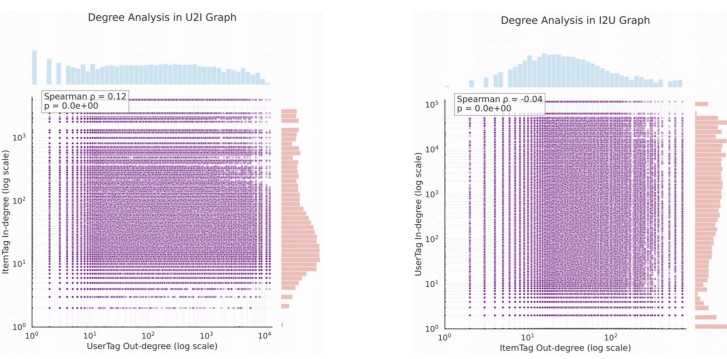

Figure 12: The degree analysis in U2I and I2U graph.

collection, introducing three novel semantic dimensions. Correspondingly, the expanded recommendation list incorporates fresh short videos aligned with these novel tags, ultimately delivering an innovative user experience through logic discovery.

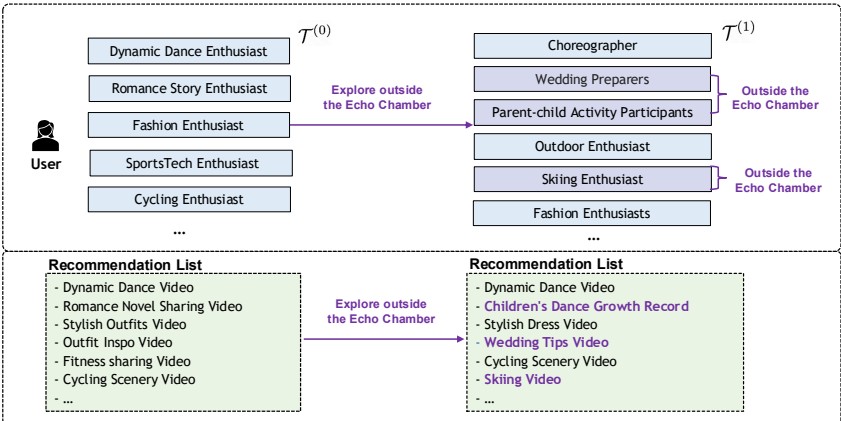

Figure 13: Case study on a user's original (user)tags and exploration (user)tags during inference.

# D   Additional Evaluation Results

## D.1   (M)LLM Evaluation with Human Experts

Table 7: Generated tag comparison results against GPT-4o.

| Test Set | (G+S)/(B+S) | (G+S)/(B+S) 95% CI | Win-Tie Rate | G/S/B Details |
|---|---|---|---|---|
| 359 videos | 0.92 | [0.993,1.35] | 59.88% | 125/90/144 |

Although multimodal large models have demonstrated strong capabilities in content understanding and reasoning for short videos, they may still suffer from hallucination issues at this stage. To validate the quality of the tags extracted by the MLLMs, we conducted a manual GSB Evaluation (Good Same Bad) [26] and a a fine-grained evaluation to assess the quality of the tags generated by the MLLMs. This manual evaluation was performed by trained professionals who systematically scored each output tag against predefined criteria. Specifically, we selected a test set of 359 short videos and compared the fine-grained scores of tags generated by our method with those generated by GPT-4o [28]. The human evaluation consists of two parts: a GSB assessment on the full set of 359 test samples (shown in Table 7) and a fine-grained evaluation on a subset of 191 samples (shown in Table 8). The fine-grained criteria include four dimensions: **Accuracy**, **Completeness**, **Reasonableness**, and **Interpretability**.

- **Accuracy**: Evaluates whether the model's output contains errors—for example, extracting tags from the video title or image OCR that are completely unrelated to the video content (ignoring weak relevance; only considering obviously incorrect labels).

- **Completeness**: Assesses whether the model's tags cover all key aspects of the video content—i.e., whether any important dimension is missing.

- **Reasonableness**: Refers to cases where tags are not outright wrong but are only weakly related to the video's main theme (e.g., mentioning incidental or background elements).

- **Interpretability**: Measures whether the tags are easy to understand, using clear and concise language while avoiding vague or obscure expressions.

Table 8: Fine-grained tag quality comparison

| Test Set | Model | Accuracy | Completeness | Reasonableness | Interpretability |
|---|---|---|---|---|---|
| 191 videos | V1 | 0.88 | 0.65 | 0.93 | 0.99 |
| 191 videos | GPT-4o | 0.85 | 0.75 | 0.92 | 0.99 |

The human evaluation results demonstrate that in terms of overall effectiveness, our method achieves a GSB score of 0.92 compared to GPT-4o. At the fine-grained level, our approach outperforms GPT-4o in accuracy, shows slightly lower performance in completeness, and performs marginally better in reasonableness. These results substantiate the superior quality of the tags extracted by our method.

Note: Compared with tag extraction, we keep a higher tolerance for the factual accuracy of the generated logic graphs, as their primary objective is to facilitate user interest exploration. This goal prioritizes diversity and the stimulation of potential user interests over strict factual precision. Nevertheless, to objectively assess the quality of these graphs, we conducted a corresponding human evaluation study. The results on a test set of 3,220 videos are summarized in the table 9.

Table 9: Tag logic graph comparison results against GPT-4o.

| Test Set | (G+S)/(B+S) | (G+S)/(B+S) 95% CI | Win-Tie Rate | G/S/B Details |
|---|---|---|---|---|
| 3,220 videos | 0.875 | [0.955, 1.19] | 52.3% | 1237/500/1483 |

## D.2 Different LLM Size & Complexity

**Different LLM Size:** To determine the optimal LLM size for our *TagCF* framework in an industrial setting, we conducted extensive experiments with LLMs of various parameter scales, including 0.5B, 1B, 7B, and 9B versions. The key findings from our scaling study are summarized in the table 10. Our parameter scaling experiments found that while smaller models (0.5B/1B) handle 93% of cases, the 7B model is crucial for the hardest 7%. The 9B model offered only a marginal +3% accuracy gain but with significantly higher latency. Thus, we employ a cost-effective cascade of smaller models for easy cases and the 7B model for hard samples, achieving an optimal balance.

**Complexity:** To optimize computational efficiency, our system performs tag extraction in a threshold-based manner, processing only new videos that exceed a predefined interaction count (e.g., 500 interactions). This selective approach ensures that resources are allocated to higher-impact content while maintaining tagging quality. Furthermore, model distillation is employed to enhance inference efficiency, enabling the distilled model to extend coverage to all videos cost-effectively. For online integration, we leverage a highly efficient key-value (KV) database, which allows for the retrieval of a video's associated tag set in constant $\mathcal{O}(1)$ time complexity. Our online workflow is designed to facilitate parallel reading of tags and subsequent modeling computations. Crucially, once extracted, the tags are stored as immutable metadata permanently linked to the video. This persistent tag set serves all downstream recommendation tasks throughout the video's lifecycle, significantly enhancing the overall performance and reusability within the system.

Table 10: Different LLM Size Results (7B as Baseline)

| Model | Accuracy | Coverage | Hard Case | Relative Cost |
|---|---|---|---|---|
| 0.5B / 1B | -7% | -7% | × | -69% |
| 7B | - | - | ✓ | - |
| 9B | +3% | +0% | ✓ | +37% |

# E   Broader Impacts

Our work on enhancing recommender systems through LLM-enhanced user role identification and logical Recommendation has significant societal implications, both positive and negative. By incorporating user roles and behavioral logic, our framework enables more nuanced recommendations, better aligning with individual preferences and social contexts. This can enhance user engagement and satisfaction in applications such as e-commerce, content platforms, and educational tools. On the other hand, the framework may potentially provide new methodologies to social science by providing automatic and systematic solutions to discover user behavioral logic in the big data era.

However, despite the advancements offered by our method, it is essential to acknowledge potential drawbacks. If the system misinterprets user roles or behavioral logic, it could lead to irrelevant or harmful recommendations. Additionally, concerns regarding privacy and fairness arise due to the collection and analysis of user data for recommendations, necessitating careful consideration of ethical implications in its deployment. To this end, further complementary research on the solutions to mitigate these issues is necessary to achieve a benign and protective recommender system for users.

# F   Limitations and Future Work

**Deal with cold start users:** In this work, we focus on a standard top-N recommendation task that assumes the presence of user histories. The proposed *TagCF* also involves a tag-based user encoder that uses a sequential model backbone. Thus, in cold-start user scenarios, where the users provide little information about their preferences, it would be difficult to solve the user role identification task or to investigate which logic the user follows.

**Improving expressiveness of the tag set:** *TagCF* can obtain a sufficiently expressive and general tag-logic knowledge that can transfer to other tasks or augmentation models. Yet, we are skeptical about the optimality of the extracted knowledge, mainly due to the greedy cover set update algorithm.

**Computational cost:** All three modules in our proposed framework brings extra computational overheads to the system. The tag extraction module and the logic reasoning module involves the inference cost of MLLMs and LLMs. However, due to the generalizability of this tag-logic knowledge, they can benefit many other task across the platform. This is also one of the key reason the augmentation paradigm of LLM-based recommender system are most favored in recent days. On the other hand, the tag-logic integration module requires extra efforts to model the tag-based encoder, learn additional objective, and explore the tag-logic during inference. These are all inevitable computational costs that the designer have to consider when constructing cost-effective solutions.

**Full tag set vs. cover set:** For efficiency and generalizability concerns, *TagCF* adopt the cover sets for tag-logic representation and augmentation of recommender systems. However, the cover set only takes a small portion of the full set, which leaves the majority of the full set knowledge unused. Intuitively, it is reasonable to believe that the more fine-grained full tag set may potentially have better interpretability for specific cases, and it may work investigation on better ways to exploit this full set.

