# OpenReview forum: "Who You Are Matters: Bridging Interests and Social Roles via LLM-Enhanced Logic Recommendation"
_NeurIPS.cc/2025/Conference — NeurIPS 2025 poster_

### Official Review · Reviewer_GKEQ · 2025-07-01

**Clarity:** 3
**Significance:** 3
**Originality:** 3
**Rating:** 5
**Confidence:** 4

**Summary:**

This paper introduces TagCF, a novel framework for recommender systems that aims to move beyond traditional item-correlation models. The authors argue that mainstream approaches neglect the crucial context of a user’s real-world social roles (e.g., “new parent,” “musician,” “student”), which are often the logical “confounders” explaining their interests. TagCF addresses this by explicitly modeling user roles and the logical relationships between roles and item topics.

The framework leverages Large Language Models (LLMs) in a multi-stage process:

- Tag Identification: A Multi-modal LLM (MLLM) analyzes items (e.g., videos) to automatically extract two types of descriptive tags: item-topic tags (e.g., “Folk Music Shows”) and user-role tags (e.g., “Folk Music Lover”).
- Logic Modeling: A separate LLM infers the logical connections between these user and item tags, creating a “collaborative logic graph.” This graph models relationships like “a Folk Music Lover is likely interested in Instrument Shows” (U2I logic) and “an item about Bow Instruments is relevant to a Symphonist” (I2U logic). To handle the massive scale, the system uses techniques like cover set reduction and model distillation.

This extracted tag-logic knowledge is then integrated into a standard sequential recommender system to enhance performance. The integration happens through tag-based encoders, a contrastive learning objective, and a logic-based scoring extension during inference. The authors demonstrate the effectiveness of this approach through extensive offline experiments on public and industrial datasets, and critically, through a large-scale online A/B test, showing significant improvements in user interaction and recommendation diversity.

**Questions:**

- Tag Pool Generation and Maintenance: The paper describes a dynamic cover set reduction algorithm (Alg. 1) to maintain a stable and expressive set of tags (T*, C*). However, the process still seems complex. Could you clarify how the initial, unrestricted tag sets (T, C) are managed? Do they grow indefinitely, or are there pruning mechanisms beyond the D-day history check in the cover set? How sensitive is the final recommendation performance to the cover set threshold τ and the history window D?
- User Role vs. Item Topic Stability: A key claim is that “user role tags are empirically more stable and more expressive in representing personalities” than item tags. This is supported by the smaller cover set size (Table 6) and superior accuracy of TagCF-ut (Table 2). Could you elaborate on why you believe this is the case? Is it because user roles are more fundamental and timeless concepts (e.g., “parent,” “student”) compared to item topics which can be ephemeral trends (e.g., a specific meme or news event)? A deeper discussion on this point would strengthen this important finding.
- Baseline Comparison and Feature Ablation: The online baseline is described as a “state-of-the-art ranking system.” It is crucial to know if this baseline already incorporates some form of user role or persona modeling, even if manually defined (e.g., using user profile data or clustered behavior). If so, your method’s improvement is even more impressive. If not, could the performance gain be partially attributed simply to adding any form of user role information, rather than your specific LLM-based approach? An ablation that compares TagCF-ut to a simpler baseline using predefined user roles would be very insightful.

**Ethical Concerns:**

["NO or VERY MINOR ethics concerns only"]

**Final Justification:**

I will keep the score to be the same and really appreciate the efforts the authors put into the rebuttal. I just read them all.

**Limitations:**

Yes, the authors have adequately addressed the limitations and potential negative societal impacts of their work in Sections F and G. They have been upfront about several key challenges:

- Cold-start users: They correctly identify that the user-history-dependent model struggles with new users.
- Optimality of the tag set: They acknowledge the greedy nature of the cover set algorithm might not be optimal.
- Computational cost: They are transparent about the significant overhead introduced by the framework.
- Broader Impacts: They responsibly discuss the dual-edged nature of their work—improved personalization vs. potential for misinterpretation, harmful recommendations, and privacy concerns.

One critical point that could be added or emphasized is the dependency on the specific MLLM/LLM chosen. The quality and nature of the extracted tags and logic graphs are entirely dependent on the underlying “world knowledge” and potential biases of the base models (M3, Qwen2.5). A different LLM might produce a completely different logic graph, leading to different recommendation outcomes. This reliance on proprietary or rapidly evolving models is a key limitation of this entire line of research.

**Paper Formatting Concerns:**

NO or VERY MINOR formatting concerns only

**Quality:**

3

**Strengths And Weaknesses:**

Strengths:

- Quality & Significance: The paper tackles a significant and well-motivated problem: moving beyond simple correlational models to incorporate deeper, more interpretable “logical” reasoning into recommendations. The proposed tasks of “user role identification” and “behavioral logic modeling” are a valuable conceptual contribution. The work is particularly strong due to its extensive and convincing empirical validation, including a live A/B test on a massive industrial platform, which is a rare and high bar for academic papers.
- Originality: The core idea of using LLMs to explicitly model a bipartite graph of user roles and item topics, and then reasoning over this graph to improve recommendations, is highly original. While other works use LLMs for recommendations, TagCF’s focus on identifying latent user roles as the logical bridge between items is a novel and powerful paradigm. The practical solutions for industrial-scale deployment, such as cover set reduction and model distillation, add to the paper’s originality and practical value.
- Clarity: The paper is well-written and structured. The motivation is clearly articulated with an intuitive example (Figure 1), and the proposed TagCF framework is broken down into logical, well-explained components (Figure 2). The authors do a good job of explaining both the high-level concepts and the low-level implementation details.

Weaknesses:

- Clarity on the role of LLMs: The paper states it uses a pre-trained MLLM (M3) and an LLM (Qwen2.5) “without finetuning” to preserve their world knowledge. However, it then immediately introduces distilled models (Pθ for tag extraction and Pφ for logic reasoning) that are trained on the LLM outputs. This could be clarified. The key weakness is the complexity and potential for information loss in this multi-stage distillation process. The performance is highly dependent on how well these smaller, specialized models can capture the knowledge of the giant, generalist LLMs.
- Complexity: The overall system is quite complex, involving multiple LLM/MLLM calls, two types of tag sets (full vs. cover), daily updates, model distillation, and three separate integration modules into a standard recommender. While the results justify the complexity, it represents a significant engineering and computational overhead, which might be a barrier to adoption for others.
- Analysis of Failure Cases: The paper presents strong positive results but would benefit from a deeper analysis of the system’s failure modes. When does the LLM-based tag extraction fail or produce “hallucinated” tags? When does the logic graph lead to poor or nonsensical recommendations? Discussing these limitations would make the work more complete.

---

> ### Author Rebuttal · Authors · 2025-07-30
>
> We thank the reviewer for the positive feedback and constructive suggestions and we address the comments as follows:
>
> 1. ***Clarity on the role of LLMs***:
> - In our method, only the distilled models $P_\theta$ and $P_\phi$ are trained, while the (M)LLMs (M3/Qwen2.5) remain frozen (see Lines 118-125, 605-623). As a result, we do not include parameter $\theta$ and $\phi$ in Eq.1 and Eq.2.
> - We'll clarify this architecture separation more prominently to address this confusion.
>
> 2. ***Complexity***:
> - To optimize processing efficiency, we selectively perform tag extraction only on new videos that exceed 500 interactions. This threshold-based approach ensures that computational resources are allocated to higher-impact content while maintaining tagging quality. Additionally, model distillation further enhances the inference efficiency of our method and it is able to cover all videos. For the integration modules, we employ a highly efficient KV (Key-Value) query database in our online system, enabling us to retrieve the tag sets corresponding to videos with $O(1)$ time complexity. Our online workflow also facilitates parallel tag reading and modeling. Importantly, the extracted tags become immutable metadata permanently associated with the videos, serving all downstream tasks throughout their lifecycle and significantly enhancing overall system performance.
> - Though the distillation process introduces new stage into the system, we remind readers that there are two major purposes: 1) It reduces the tag information to a smaller, more stable, and more general coverset that better suits for downstream tasks including but limited to recommendation; 2) In systems with a large number of new items uploaded each day, it reduces the workload of MLLMs/LLMs. Conversely, this distillation process becomes unnecessary if the target tag set is small and given to the LLM and there is not many new items to process each day.
> - We will provide a more detailed discussion of potential computational costs in the revised version of the paper.
>
> 3. ***Accuracy of distilled model***:
> - To mitigate the information loss of distillation, we ensure that the distilled models maintain a high level of accuracy by selecting the top-k predictions for video-to-tag mapping and top-b predictions for tag-to-tag mapping, typically k=10 and b=10 in our implementation. Empirically, the models can effectively fulfill the performance requirements. Additionally, by increasing or decreasing k or b, the distilled models introduce a trade-off between accuracy and inference speed.
> - To provide further insight into this balance, we will include detailed accuracy/speed trade-off curves in the Appendix.
>
> 4. ***Failure Cases***:
> - We acknowledge the existence of possible hallucination cases. In Table 7, we provide a manual GSB evaluation, and in Table 8, we offer a fine-grained tag quality comparison between the tags generated by our system and those generated by GPT-4o. Though the accuracy is sufficient for production use, it is still not 100%. Fortunately, when the LLM produces "hallucinated" tags, it may not necessary affects the system's performance in a bad way, since the tag is still related to the video while being surprisingly different, which potentially improves the diversity. Still, one should try to minimize this error (e.g. by removing tags that contain incorrect information) to ensure recommendation accuracy.
> - In contrast, the purpose of the logic graph is to explore user interests, so a certain degree of accuracy loss is permissible. We also investigate the quality of the logic graph by analyzing the edge degree of the U2I and I2U logic graph in Figure 12. To be more rigorous, we conducted a manual GSB evaluation to compare the accuracy of logic graphs generated by our large model versus GPT-4o, specifically assessing whether inferred tag relationships were erroneous (e.g., cases where inferred tags were semantically or factually irrelevant to original tags). The evaluation was performed on a sample set of 3,220 videos, and we will add the following quantitative results to our paper to enhance the completeness of our work.
> | Test Set       | (G+S)/(B+S) Ratio | 95% CI    | Win-Tie Rate | G/S/B Distribution (Counts) |
> |:-----------:|:------------:|:---------------:|:--------------:|:-----------------------:|
> | 3,220 videos   | 0.875             | [0.95, 1.19] | 52.3%        | G:1,237 / S:500 / B:1,483   |
>
> 5. ***Tag pool generation and maintenance***:
> - So far, we do not restrict the full tag sets (T, C) and only apply the D-day check on cover sets. In the full sets, we have observed a decreasing number of updates, meaning that the storage may eventually converge some day in the future, though in a much slower rate than the cover sets. Still, the size of these full sets is statistically much smaller than the number of videos.
> - Additionally, in practice, we find that tags follow an extremely skewed frequency distribution (Figure 11), indicating that not all tags are identically useful and expressive. This motivates our proposal of the dynamic cover set reduction algorithm (Algorithm 1) to maintain a stable and expressive set of tags (T*, C*). In terms of the removal mechanism, we only considered the D-day recall check but does not rule our the possibility of including other mechanisms in the future.
> - We will provide more explanations for this algorithm in our revised paper to improve methodological clarity.
>
> 6. ***Sensitivity of $\tau$ and $D$***:
> - These parameters are primarily determined by industrial requirements for all businesses rather than being tuned based on recommendation outcomes.
> - The threshold τ is designed to cover 99% of the videos, which is a critical industrial requirement. Achieving such a high coverage rate is essential for meaningful online application, and covering smaller number of videos would lead to significant information loss, making the system ineffective for business use.
> - The history window D is dictated by storage and resource consumption constraints. While different settings can be applied, a smaller history window increases dynamism, leading to instability in the tag sets. Setting a longer D helps improve tag stability and enhances recommendation performance.
> - We will address these concerns in our revised paper.
>
> 7. ***User role vs. item topic stability***:
> - Our claim about the superior stability and expressiveness of user role tags is supported by empirical observations: In Table 6, we present the statistics of the full user/item tag set and the reduced user/item tag cover set. It can be seen that the daily addition of full-size item tags is 3.8 million, while the daily addition of user tags is 0.26 million, indicating that the collection of user tags is more stable than that of item tags. For coversets, item tag's daily removal is around hundreds, but the user tag have almost converged and may never update for days. Intuitively, a smaller set with less frequent removal usually indicates a more stable tag set with longer lifespan. We will add this statistics in Table 6 as additional support and provide additional discussion of this point in the appendix.
> - We consider "user roles are more fundamental and timeless concepts compared to item topics which can be ephemeral trends" as the possible reason for this belief, but whether this is the true and only reason may require further investigation. So far, we choose to use the aforementioned empirical statistical results as verification.
>
> 8. ***Baseline comparison and feature ablation***:
> - Yes, our online baseline has already incorporated user profile data (such as user’s age, gender, etc.) with PPNET[1] and several manually designed tag features for videos similar to the related works mentioned in the paper. Note that the video-to-tag and tag-logic graph generation are merely based on the ''world knowledge'' of MLLMs/LLMs, which does not require the participation of the recommender system. This could be a potential advantage since it may provide signals and patterns that are useful but unseen in the existing solution, boosting the performance.
> - We will include this detail in the revised version of the paper.
> [1] PEPNet: Parameter and Embedding Personalized Network for Infusing with Personalized Prior Information. (KDD '23).

---

### Official Review · Reviewer_xZHV · 2025-07-02

**Clarity:** 3
**Significance:** 2
**Originality:** 3
**Rating:** 4
**Confidence:** 5

**Summary:**

The paper introduces a novel recommendation framework called TagCF, which aims to enhance the performance of recommender systems by explicitly modeling user roles and the logical relationships between user roles and item topics. Traditional recommender systems often overlook the importance of user characteristics and their social roles, focusing primarily on item topics and user-item interactions. This paper addresses this gap by proposing a solution that integrates Large Language Models to extract user and item tags and infer logical connections between them.

**Questions:**

Please refer to Weakness seciton.

**Ethical Concerns:**

["NO or VERY MINOR ethics concerns only"]

**Limitations:**

This study may raise some fairness issues, such as LLM-inferred user roles (e.g., "low-income parents") could propagate demographic biases.

**Paper Formatting Concerns:**

None observed.

**Quality:**

3

**Strengths And Weaknesses:**

Strengths:
1. The paper proposes an innovative framework, TagCF, which effectively addresses the limitations of traditional recommendation systems by incorporating multi-modal large language models (MLLMs) and recommendation systems, thus capturing user roles and social roles.
2. The experimental approach is sound, with access to both industrial data and easily accessible public datasets. The effectiveness of the framework has been validated in the industrial setting, and the ablation studies and hyperparameter analyses are quite comprehensive.
3. The paper elaborates on each component of the TagCF framework and its working mechanism, including MLLM-based tag extraction, LLM-based collaborative logic filtering, and the integration of tag logic into recommendation systems.

Weaknesses:
1. The paper utilizes the reasoning ability of LLMs in constructing logic graphs, it does not provide detailed solutions for ensuring the accuracy and consistency of the logic graphs, or for handling logical conflicts and contradictions.
2. The paper mentioned that Qwen2.5-7B was used to iteratively infer the U2I and I2U logic. Have you tried to use models with other parameter values, such as 3B or 9B, to explore the impact of different large model parameter values ​​on reasoning performance and the robustness of the entire framework?
3. The experiments on public datasets only selected Books and Movies, while the industrial application is based on video recommendations. It is suggested to supplement the experiments with two public video recommendation datasets, Microlens and PixelRec, to further demonstrate the effectiveness of the framework in video recommendations.
4. It is suggested to present a case study to illustrate the “Dynamic Cover Set Reduction Algorithm” mentioned in the paper.
5. Daily MLLM processing of 3M videos requires 50× NVIDIA 4090 GPUs (§C.1). While distillation mitigates this, cold-start tag extraction for new items remains costly.
6. Logic-based tag expansion involves graph traversals – latency impact on ranking stage is unquantified.
7. Are there fairness risks? LLM-inferred user roles (e.g., "low-income parents") could propagate demographic biases.

---

> ### Author Rebuttal · Authors · 2025-07-30
>
> We thank the reviewer for the positive feedback and constructive suggestions and we address the comments as follows:
>
> 1. ***Accuracy and consistency of graphs***:
> As we have illustrated in Appendix E, we validated the correctness of the generated tags of videos with both human evaluation and larger model (i.e. GPT-4o) comparison. **In contrast, we give more tolerance to the correctness of graphs, since the main purpose of these logic graphs are interest exploration, which is more diversity oriented rather than accuracy oriented.** To be more rigorous, we also conduct expert human assessments on the logic graphs generated by our LLM, with comparative analysis against GPT-4o's reasoning outputs. We conducted a manual GSB evaluation (Good Same Bad)  to compare the accuracy of logic graphs generated by our large model versus GPT-4o, specifically assessing whether inferred tag relationships were erroneous (e.g., cases where inferred tags were semantically or factually irrelevant to original tags). As a clarification, we will add the following quantitative results to our paper to enhance the completeness of our work.
> | Test Set    | (G+S)/(B+S) Ratio | 95% CI     | Win-Tie Rate | G/S/B Distribution (Counts) |
> |:------------:|:------------------:|:-----------:|:-------------:|:---------------------------:|
> | 3,220 videos| 0.875             | [0.95, 1.19] | 52.3%      | G:1,237 / S:500 / B:1,483   |
>
> 2. ***Different LLM size***:
> Yes, we extensively tested various parameter sizes  (including 0.5B, 1B and 9B versions). We found that smaller models (0.5B/1B) can achieve reasonable accuracy for 93% of cases, while the 7B variant remains essential for resolving the most challenging 7% of samples where smaller models fail. Although the larger 9B model provided a slight improvement in accuracy (+3%), it also increased inference latency, making 0.5~7B the optimal trade-off. To reduce costs, the industrial setting uses smaller 0.5B/1B models for the first generation, then uses the 7B to resolve the hard samples. The resulting mechanism reduces operational costs by 37% compared to the 9B model, offers 98% video coverage and maintains production-grade latency. This balance of cost and performance led us to standardize on the 7B version. We will provide detailed parameter scaling results in the appendix.
>
> 3. ***Other video recommendation datasets***:
> Due to the time limit, we can only provide a preliminary results for MicroLens-50K, a small version of one of the datasets mentioned by the reviewer. We present the results of the best baselines and our method in the following table. As shown below, our method has also demonstrated its effectiveness on this public video recommendation dataset. Note that the I2U and U2I graphs are transferrable, but one still have to run large models to generate tags to use these features.
> | model | NDCG@10 | NDCG@20 | MRR@10 | MRR@20 | Cover@10 | Cover@20 | GINI@10 | GINI@20 |
> |---|---|---|---|---|---|---|---|---|
> | Mamba4Rec |0.0328 |0.0403 |0.0215 |0.0239 |0.8037 |0.8510 |0.7796 |0.7734 |
> | SAID |0.0332 |0.0410 |0.0220 |0.0241 |0.7952 |0.8378 |0.7902 |0.7880 |
> | TagCF-it | 0.0357|0.0439|0.0236 |0.0252 |**0.8511** |**0.8804** |**0.7198** |**0.7302** |
> | TagCF-ut | **0.0368**|**0.0452** |**0.0243** |**0.0264**|0.8414 |0.8751 |0.7264 |0.7350 |
> 4. ***Dynamic coverset algorithm case study***:
> The dynamic nature is mainly driven by the continuous addition of new tags to the cover set and the removal of outdated tags. To provide a clearer understanding, we have included a case analysis in the table below and we plan to incorporate both the table and a detailed case analysis in the revision phase. The table presents the statistics and adjustments of the cover set after applying the Dynamic Cover Set Reduction Algorithm.
> | Cover Set Statistics | Results | Description |
> |---|---|---|
> |Previous Video Coverage Rate | 99.00% | This indicates the coverage rate of the cover set before applying the algorithm|
> | Previous Video Recall Rate | 80.34%  | This indicates the recall rate of the cover set before applying the algorithm |
> | Updated Video Coverage Rate | 99.39% | After optimization by the algorithm, the coverage rate of the cover set slightly improved|
> | Updated Video Recall Rate | 98.61% | After optimization by the algorithm, the recall rate of the cover set significantly improved |
> | **Cover Set Dynamics** | **Results** | **Description** |
> | Outdated Tag Count | 182 | These are the tags removed from the cover set because they did not appear in the historical records |
> | Remaining Tag Count | 22,757 | This is the number of tags retained in the cover set after optimization by the algorithm.
>
> 5. ***Cold start tag extraction cost***:
> For new videos, their multi-modal information will first generate the embedding through M3 (described in section 2.1). Then the distilled model will use the this embedding to infer corresponding tags in the cover set, which is very efficient in practice and it is able to cover all videos. If you are referring to the tag extraction with pure MLLM in the full open language space using Eq.1, then there do exists a computational bottleneck. To circumvent this in practice, we perform pure MLLM-based tag extraction only on videos exceeding 500 interactions, rather than applying indiscriminate extraction across all uploads. This threshold-based approach ensures computational resources are allocated to higher-impact content while maintaining tagging quality. Fortunately, we only need to run this process once for each video, and the tag features will serve all downstream tasks throughout the video's lifecycle. We will add this implementation detail in the revised manuscript.
>
> 6. ***Quantify graph traversal***:
> In our implementation, both video-to-tag and tag-logic graphs are stored as large key-value (KV) lookup tables, and the parallel lookup process ensures that the time complexity for this lookup phase is $O(1)$ (including the initial tag set and the exploration tag set). When calculating the tag-based scores, the overall traversal complexity is $O(b^2 k)$ (see Lines 665-668), which is the same as the size of $\mathcal{T}(1)$, and one can boost this process using parallel computing (mentioned in Appendix B.6). During the model inference stage, tag scoring is performed in parallel with other scoring processes, which usually requires computation of more complicated models. Therefore, there is no additional latency increase in our online experiments.
>
> 7. ***Fairness issues***:
> Yes, we acknowledge the existence of the potential demographic bias propagation through LLM-inferred user roles and item tags. This is partially reflected by the power law distribution of tags in Figure 11 of Appendix D.1. In response, we will expand our limitations section with the discussion on potential fairness issues. We thank the reviewer for bringing this perspective which significantly improves our work's completeness.

---

### Official Review · Reviewer_vTUY · 2025-07-02

**Clarity:** 3
**Significance:** 3
**Originality:** 2
**Rating:** 4
**Confidence:** 3

**Summary:**

This paper introduces the user role identification task and behavioral logic modeling task through an integration framework of (multi-modal) large language models. In addition, taking advantage of the extracted tag-logic information can further augment the recommendation performance. Experiments conducted both online and offline demonstrate the effectiveness of the proposed framework and empirically show that user role identification is potentially a good choice.

**Questions:**

1. Why the logic reasoning only depend on the open-world knowledge of LLM, ignoring the important collaborative filtering in training data?
2. In Eq (4), why use ID-based item embedding X_i as the representation of the item instead of tag-based item encoding or the combination of both ID-based and tag-based item embedding?
3. Why tag-based learning augmentation can benefit recommendation performance? Is there any more in-depth discussion or analysis?

**Ethical Concerns:**

["NO or VERY MINOR ethics concerns only"]

**Final Justification:**

I have read the authors' rebuttal and keep my original rating.

**Limitations:**

The authors have discussed most of limitations of their work, but lacking discussion on user privacy concerns.

**Quality:**

2

**Strengths And Weaknesses:**

Strengths:
1. The evaluation of the real-world online scenario demonstrates significant credibility to the practical value of the proposed framework.
2. Various experiments are conducted, including detailed hyperparameter analysis, ablation studies, and comparisons across multiple baselines.
3. The paper is well-organized and easy to follow.

Weaknesses:
1. The motivation for incorporating the user role identification task needs more in-depth discussion or theoretical support. For example, whether the traditional ID-based or profile-based user embeddings already implicitly contain user role information? Why is it necessary to explicitly identify user roles?
2. Some experimental phenomena lack deeper analysis. For instance, why do TagCF-util and TagCF-expl perform significantly differently in terms of interaction and diversity? It would be better to provide some more discussions. In addition, in lines 312-313, this paper claims that "item tags may have a shorter lifespan and may frequently update even in the cover set". Detailed support should be provided for this claim.
3. The introduction of user role identification may raise user privacy concerns, which is particularly critical in privacy-sensitive scenarios.

---

> ### Author Rebuttal · Authors · 2025-07-30
>
> We thank the reviewer for the positive feedback and constructive suggestions and we address the comments as follows:
> 1. ***Motivation of user role identification***:
> - We used to have the same concern as the reviewer's, and that is also the reason we conducted various rigorous analysis including the statistical comparison in section 3.2.5 and Appendix D.1, as well as the comparison in the main result of Table 1. Based on all these empirical results, we believe that we have provided an evidence that user role identification and tag-logic extraction might indeed be an missing part of the current recommender system design.
> - Intuitively, although ID-based and profile-based user embeddings may implicitly contain user role information, they still exhibit certain limitations including but limited to inaccurate modeling of user behaviors (as the examples shown in introduction), insufficient interpretability from black-box solutions, and lack of explicit strategic control over the recommendation policy (compared to the tag-logic inference extension). As we have mentioned in the paper, some interpretability-oriented solutions[1,2] may partially address these limitations, but most existing approaches rely on confined feature engineering within closed domains, resulting in limited real-world applicability and lacking of generalizability.
> - Empirically, in Table 2, we compared against ID-based (Mamba4Rec, the state-of-the-art in this category) and profile-based user embedding methods (RLMRec which leverages LLMs for user representation). While these baselines demonstrate competent performance, they remain suboptimal compared to our method. In general, we believe that comprehensive role identification and behavior logic discovery may complement the current ID-based recommender system.
> - Additionally, in industrial perspective, the user role knowledge and user behavior logic graphs are originated from recommendation data but potentially has broader impact for all video producers, various businesses or even social study.
> In general, we sincerely appreciate the insightful comments and we will integrate this motivation discussion more explicitly into the revised manuscript, particularly in Introduction and Method Section to strengthen the justification for our role-driven approach.
> [1] Explainable recommendation with fusion of aspect information, WWW, 2018
> [2] Triple Dual Learning for Opinion-Based Explainable Recommendation, TOIS, 2023
>
> 2. ***TagCF-util vs TagCF-expl***:
> Intuitively, the initial tag set comes from the user's past behavior so it best aligns with the user's ``echo chamber'' (or the observed or presented interest by user), while the TagCf-expl uses the logic graph to extend these tags to a new set of tags which is likely to be different from the initial set, achieving better diversity during recommendation. Due to the limited space, we have provided a case study to better illustrate this differences in Appendix D.3 instead of the main body of the paper.
>
> 3. ***Support for shorter item tag lifespan***:
> The full set continuously expand without removal, and the differences in the daily expansion number in Table 6 can partially supports this claim. For coversets, item tag's daily removal is around hundreds, but the user tag have almost converged and may never update for days. Intuitively, a smaller set with less frequent removal usually indicates a more stable tag set with longer lifespan. We will add this statistics in Table 6 as additional support and provide additional discussion of this point in the appendix.
>
> 4. ***CF data vs. open-world reasoning***:
> We intentionally make the tag-logic extraction process independent of collaborative data to better discover real-world knowledge that enhances recommendation but are not typical in the collaborative filtering data. This is also why we consider the graphs as general knowledge that are transferable to other datasets. Still, we do not rule out the possibility of using collaborative filtering data for logical reasoning, so your question also raises a key future direction that compares different knowledge extraction methodologies for the TagCF framework: end-to-end logic generation with CF vs. LLM-based logic + CF alignment. Thanks for your attention and suggestions.
>
> 5. ***ID Embedding in Eq.4***:
> In lines 649-651, we have mentioned that any feature encoding fusion method is theoretically feasible but empirically separate design achieves the best result. Yet, we do not observe significant difference for different embedding schemes for the target item embedding in Eq.4, so we keep the original design that uses the ID embedding since we are calculating the raw score. We previously experimented with tag-based item encoding and the combination of both ID-based and tag-based item embedding, but the results ultimately showed that this fusion method is the best.
>
> 6. ***Why tag-based learning***:
> Tag-based Learning augmentation helps in the alignment of embedding spaces by ensuring that the user and item embeddings are consistent with the tag embeddings. While we have provided empirical proofs for this design, intuitively, we believe that this might be related to the generalizability for tag-based contrastive learning compared with noisy item-based learning. Nevertheless, the fundamental reasons for this effectiveness may require further investigation in the future.
>
> 7. ***Privacy issues***:
> In our current system design, we consider the recommender trustworthy and benign to users, but we agree with the reviewer that this might not always be the case when users do not trust or partially trust the system. We believe that the privacy issue could be one of the key future direction for TagCF, and standard solutions like encryption methods or adversarial methods may all need specialized designs. We will include this discussion in the limitations section and we thank the reviewer for bringing this perspective which significantly improves our work's completeness.

---

### Comment · Area_Chair_viib · 2025-08-01

Dear Reviewers,

The responses from the authors are available. Please read them to see if you have any further question.

Thank you for your support.

AC.

---

### Author Response · Authors · 2025-08-08

Dear Reviewers and ACs,

We sincerely appreciate your time dedicated to reviewing our work and constructive efforts in the discussion. We believe your insightful comments have not only strengthened our paper but also provided valuable directions for future research in the field. Below, we summarize the key strengths acknowledged by the reviewers and highlight how we addressed their concerns in our revisions.

***Strengths***: Credible evaluation (Reviewer vTUY, xZHV, GKEQ), well organized paper and detailed model elaboration (Reviewer vTUY, xZHV, GKEQ), and innovative TagCF framework with significant impact (Reviewer xZHV, GKEQ).

***Addressed Concerns with Responses***: LLM Scalability & Cost (Reviewer vTUV, xZHV, GKEQ), verification of the extracted graph knowledge (Reviewer xZHV, GKEQ), justification for role-driven design (Reviewer vTUY), more discussion on hyperparameters and dynamics of cover set algorithm (xZHV, GKEQ), and other presentation add-ups.

***Extended Discussion on Limitation and Future Directions***: Privacy, fairness, and hallucination concerns (Reviewer vTUV, xZHV, GKEQ)

To our esteemed Reviewers:
We are deeply grateful for your evaluation and invaluable feedback, and we have carefully addressed each of your concerns in our rebuttal and hope our responses adequately answers your questions.

To our dedicated ACs:
We sincerely appreciate the AC's assistance in reminding the reviewers of our responses and enhancements. Since the deadline for author-reviewer discussion is approaching now, we sincerely hope that our efforts in the rebuttal discussion can be taken into consideration.

Best regards to all,

Authors

---

### Note · Authors · 2025-08-15

Dear Area Chair and Reviewers,

Thanks for your time, effort, and valuable insights! To facilitate your discussion, we briefly summarize the strengths of our work and resolved issues.

***Strengths***:

* **Innovative TagCF framework** that explicitly model the bipartite graph between user roles and item topics, and **significant impact** that address the new tasks of "user role identification" and "behavioral logic modeling" (Reviewer xZHV, GKEQ).
* **Credible evaluation** with fair offline comparison, online experiments, ablation studies, and various hyperparameter analysis (Reviewer vTUY, xZHV, GKEQ);
* **Well organized and easy-to-follow paper** with detailed model elaboration (Reviewer vTUY, xZHV, GKEQ);

***Addressed Concerns***:

* **LLM scalability & cost** (Reviewer vTUV, xZHV, GKEQ)
* **Verification of the extracted graph knowledge** (Reviewer xZHV, GKEQ)
* **Justification for role-driven design** (Reviewer vTUY)
* **Extended discussion on hyperparameters and cover set algorithm**(xZHV, GKEQ)
* **More discussion on limitation and future directions**: Privacy, fairness, and hallucination concerns (Reviewer vTUV, xZHV, GKEQ).

We hope that our clarifications will help convey the value and robustness of our contributions. In general, we believe that our work opens a new direction and makes an important step towards understanding the logical connections between item topics and user roles in recommendation, and the innovative TagCF framework is verifiably effective in both offline and online scenarios. We will take all the reviewers' suggestions and improve the corresponding discussions in our paper. Again, thanks for all your constructive support and valuable insights!

Best regards,

Authors.

---

### Decision · Program_Chairs · 2025-09-17

**Decision:**

Accept (poster)

**Comment:**

This paper presents TagCF, a unified framework that augments recommender systems by explicitly modeling users’ real-world social roles and the logical relationships between these roles and item topics. Leveraging (multi-modal) large language models, TagCF extracts user and item tags and infers causal “confounders” such as “new parent” or “musician,” which traditional item-correlation methods overlook. Extensive online and offline experiments demonstrate consistent performance gains, confirming that user role identification is both effective and practically valuable for next-generation recommendation.

**Strengths**
1. To capture user role and social role, this paper proposes the TagCF framework. It involves MLLM-based tag extraction, LLM-based collaborative logic filtering, and the integration of tag logic into recommendation systems.
2. The evaluation is comprehensive. It includes both offline and online experiments. Component analysis is also covered.
3. This paper is well-written and easy to follow.

**Weaknesses**
1. The motivation and some methodologies of this paper should be better clarified. Besides, the computational cost should also be discussed.
2. The experiments could be further improved by introducing more public datasets and case studies.
3. Reviewers have concerns about the LLM scalability and privacy.

During the rebuttal stage, the authors add some experiments to clarify the data and model size issues.

After the rebuttal, all the reviewers still have positive rating scores for this paper.